# Equatorward dispersion of a high-latitude volcanic plume and its relation to the Asian summer monsoon: a case study of the Sarychev eruption in 2009

Xue Wu[1,2], Sabine Griessbach[1], Lars Hoffmann[1]

[1]Jülich Supercomputing Centre, Forschungszentrum Jülich, Jülich, Germany
[2]Key Laboratory of Middle Atmosphere and Global Environment Observation, Institute of Atmospheric Physics, Chinese Academy of Sciences, Beijing, China

*Correspondence to*: Xue Wu (xu.wu@fz-juelich.de)

**Abstract.**

Tropical volcanic eruptions have been widely studied for their significant contribution to stratospheric aerosol loading and global climate impacts, but the impact of high-latitude volcanic eruptions on the stratospheric aerosol layer is not clear and the pathway of transporting aerosol from high-latitudes to the tropical stratosphere is not well understood. In this work, we focus on the high-latitude volcano Sarychev (48.1 °N, 153.2 °E), which erupted in June 2009, and the influence of the Asian summer monsoon (ASM) on the equatorward dispersion of the volcanic plume. First, the sulfur dioxide ($SO_2$) emission time series and plume height of the Sarychev eruption are estimated with $SO_2$ observations of the Atmospheric Infrared Sounder (AIRS) and a backward trajectory approach, using the Lagrangian particle dispersion model Massive–Parallel Trajectory Calculations. Then, the transport and dispersion of the plume are simulated using the derived $SO_2$ emission time series. The transport simulations are compared with $SO_2$ observations from AIRS and validated with aerosol observations from the Michelson Interferometer for Passive Atmospheric Sounding (MIPAS). The simulations show that about 4% of the sulfur emissions were transported to the tropical stratosphere within 50 days after the beginning of the eruption, and the plume dispersed towards the tropical tropopause layer (TTL) through isentropic transport above the subtropical jet. The MPTRAC simulations and MIPAS aerosol data both show that between the potential temperature levels of 360 and 400 K, the equatorward transport was primarily driven by anticyclonic Rossby wave breaking enhanced by the ASM in boreal summer. The volcanic plume was entrained along the anticyclone flows and reached the TTL as it was transported south-westwards into the deep tropics downstream of the anticyclone. Further, the ASM anticyclone influenced the pathway of aerosols by isolating an 'aerosol hole' inside of the ASM, which was surrounded by aerosol-rich air outside. This transport barrier was best indicated using the potential vorticity gradient approach. Long-term MIPAS aerosol detections show that after entering the TTL, the aerosol from the Sarychev eruption remained in the tropical stratosphere for about 10 months and ascended slowly. The ascent speed agreed well with the ascent speed of the water vapor tape recorder. Furthermore, a hypothetical MPTRAC simulation for a wintertime eruption was carried out. It is shown that under winter circulations, the equatorward transport of the plume would be suppressed by the strong subtropical jet and weak wave breaking events. In this hypothetical scenario, a high-latitude volcanic eruption would not be able to contribute to the tropical stratospheric aerosol layer.

## 1 Introduction

The impact of volcanic aerosol on climate has received wide attention over decades. Robock (2000, 2013) gave a comprehensive review on this subject. The major sources of the stratospheric aerosol are volcanic sulfur gases, mainly in form of sulfur dioxide ($SO_2$), which are oxidized and converted into sulfate aerosol within hours to weeks (von Glasow et al., 2009). The sulfate aerosol is responsible for profound effects on the global climate (McCormick et al., 1995; Robock, 2000). Sulfate aerosol is a strong reflector of visible solar radiation and causes cooling of the troposphere; while it is also an effective absorber of near infrared solar radiation. Larger particles seen after an eruption can also absorb infrared radiation (Stenchikov et al., 1998), so they may induce heating of the stratosphere. In contrast, $H_2O$ and $CO_2$ in the volcanic emissions do not have much measureable impact on the global climate since they are less abundant than the atmospheric $H_2O$ and $CO_2$ reservoir (Gerlach, 1991; Gerlach, 2011). Volcanic ash usually only causes regional influence on climate for weeks, because of its large particle size, fast sedimentation, and consequently short life time in atmosphere (Niemeier et al., 2009).

The stratospheric aerosol layer enhanced by volcanism not only has a significant impact on the Earth's radiative budget (McCormick et al., 1995; Robock, 2000), it also has an impact on chemical processes in the lower stratosphere (Rodriguez et al., 1991; Solomon et al., 1993), in particular, on ozone depletion (Jäger and Wege, 1990; Tilmes et al., 2008; Solomon et al., 2016). Conventionally, large volcanic eruptions with a volcanic explosivity index (VEI) larger than 4, are thought to play a key role in the stratospheric sulfate aerosol budget. Recently, several studies focused on small and moderate-sized volcanic eruptions (VEI $\leq$ 4) (Kravitz et al., 2011; Solomon et al., 2011; Vernier et al., 2011; Ridley et al., 2014) and considered them as the primary source of the notable increase of stratospheric aerosol since 2000 that slowed down global warming (Solomon et al., 2011; Neely et al., 2013; Haywood et al., 2014). Models neglecting the effects of less intense volcanic eruptions tend to overestimate the tropospheric warming (Santer et al., 2014). The potential climate impact of volcanic emissions also largely depends on the time of year when the eruption takes place, the injection height and the surrounding meteorological conditions (Kravitz et al., 2011). The atmospheric consitions in different seasons influences sulfate formation and the zonal asymmetry of the polar vortex can affect the aerosol's transport. The eruption plume height is fundamental for aerosol microphysics in the atmosphere and the large-scale circulation affects the transport of the plume. So instead of only using the VEI, it is more sensible to study the atmospheric impacts by including volcanic eruptions based on the amount of stratospheric $SO_2$ injection together with eruption height (Brühl et al., 2015; Timmreck, 2012). Further information on volcanic $SO_2$ emissions and plume altitudes is crucial in climate models when estimating trends of global temperature or ozone depletion.

The majority of studies on global climate impact of volcanic eruptions focuses on tropical eruptions. Extratropical volcanic eruptions are expected to have less impact on the global climate, because the downward flow of the Brewer–Dobson circulation (BDC) in the stratospheric extra-tropics prevents sulfate aerosol from rising up (Seviour et al., 2012), and sulfate aerosol in the extratropical stratosphere easily subsides back to the troposphere within months (Holton et al., 1995; Kremser et al., 2016). Although some studies show that extratropical volcanic eruptions can also have a significant impact on climate (Highwood and Stevenson, 2003; Oman et al., 2005; Oman et al., 2006; Kravitz and Robock, 2011; Pausata et al., 2015a; Pausata et al., 2015b), the impact is usually found to be limited to the specific hemisphere in which the eruption occurred (Graf and Timmreck, 2001; Kravitz and Robock, 2011; Pausata et al., 2015b).

However, a high-latitude volcanic eruption may also contribute to the tropical stratospheric aerosol loading and even affect the other hemisphere (Schmidt et al., 2010). In this paper, we focus on a moderate volcanic eruption, the Sarychev eruption in 2009 (VEI = 4), which is considered as one of the high-latitude eruptions that affect tropical latitudes (von Savigny et al., 2015) and which is responsible for the increase in tropical stratospheric aerosol optical depth (Khaykin et al., 2017; Rieger et al., 2015). In particular, we study the transport pathway of

the Sarychev $SO_2$ emission and aerosol from the extra-tropical lower stratosphere to the tropical tropopause layer (TTL). In this study, the term "$SO_2$ emission" refers to the $SO_2$ released or injected into the atmosphere during the explosive eruption of Sarychev. The "tropical stratosphere" refers to the region between 30 °N and 30 °S above the 380 K isentropic surface. In order to analyze the transport of the Sarychev plume and to assess the role of the Asian summer monsoon (ASM) in the dispersion process, in Section 2 we first present the

Lagrangian transport model and data used for simulations. In Section 3, we derive the altitude-resolved Sarychev $SO_2$ emission time series using Atmospheric Infrared Sounder (AIRS) $SO_2$ measurements and an inverse modelling approach. The equatorward dispersion of the Sarychev $SO_2$ emissions revealed by Michelson Interferometer for Passive Atmospheric Sounding (MIPAS) aerosol detections and trajectory simulations and the relation to the ASM are shown in Section 4. Results of this study are further discussed in Section 5 and

summarized in Section 6.

## 2 Observations and model

### 2.1 AIRS

AIRS (Aumann et al., 2003) is an infrared sounder with across-track scanning capabilities aboard the National Aeronautics and Space Administration's (NASA's) Aqua satellite. Aqua was launched in May 2002 and operates

in a nearly polar, Sun-synchronous orbit at an altitude of about 710 km with a period of 98 min. The AIRS footprint size is 13.5 km×13.5 km for nadir and 41 km×21.4 km for the outermost scan angles. The along-track distance between two adjacent scans is 18 km. AIRS provides nearly continuous measurement coverage during 14.5 orbits per day and covers the globe almost twice a day. The observations provide good horizontal resolution and make it ideal data for observing the fine filamentary structures of volcanic $SO_2$ plumes.

In this study, we use the $SO_2$ index (SI, unit: K) defined in Hoffmann et al. (2014) to estimate the $SO_2$ emission time series and evaluate the Lagrangian transport simulations. The SI is defined as the brightness temperature differences in the 7.3 μm waveband.

$$SI = BT\left(\nu_{1407.2\ cm^{-1}}\right) - BT\left(\nu_{1371.5\ cm^{-1}}\right), \qquad (1)$$

where $BT(\nu)$ is the brightness temperature measured at wavenumber $\nu$. The SI increases with increasing $SO_2$

column density and it is most sensitive to $SO_2$ at altitudes from 8 to 13 km. The SI of Hoffmann et al. (2014) is more sensitive and performs better on suppressing background interfering signals than the SI provided in the AIRS operational data products. It is therefore well suited to track low $SO_2$ concentrations over time. A detection threshold of 2 K is suitable in this study to identify the Sarychev $SO_2$ emissions. More details on the SI can be found in Hoffmann et al. (2014; 2016) and Heng et al.(2016).

AIRS was able to detect the $SO_2$ cloud from the beginning of the eruption of Sarychev on 12 June 2009 up to five weeks later. Observations during the first five days after the eruption have been used for estimating the $SO_2$

emission time series. Observations at a later stage are used for comparison with the Lagrangian transport simulations.

## 2.2 MIPAS

MIPAS (Fischer et al., 2008) is an infrared limb emission spectrometer aboard European Space Agency's (ESA's) Envisat, which was in operation from July 2002 to April 2012, providing nearly 10 years of measurements. The vertical coverage of its nominal mode was 7–72 km from January 2005 to April 2012. MIPAS has a field of view of about 3 km $\times$ 30 km (vertically and horizontally) at the tangent point and the extent of the measurement volume along the line of sight is about 300 km. The horizontal distance between two

adjacent limb scans is about 500 km. In 2009, the general measurement pattern of MIPAS is to measure eight days in nominal mode followed by one day in middle atmosphere mode and one day in upper atmosphere mode. On each day, about 14 orbits with about 90 profiles per orbit are measured. From January 2005 to April 2012, the vertical sampling grid spacing between the tangent altitudes is 1.5 km in the upper troposphere and lower stratosphere (UTLS) and 3 km at altitudes above.

In this study, we use MIPAS altitude-resolved aerosol data retrieved by Griessbach et al. (2016) to compare with model results and analyze the transport of volcanic plume.

In the first step, the aerosol-cloud-index (ACI) is defined to identify cloud or aerosol contaminated spectra. The ACI is defined as the maximum value of both the cloud index (CI) and aerosol index (AI),

$$ACI = max(CI; AI), \qquad (2)$$

The CI is the standard method to detect clouds and aerosol with MIPAS (Spang et al., 2001), which is defined as the ratio between the mean radiances around the 792 cm$^{-1}$, covering a band with strong CO2 emissions and the atmospheric window region around 833 cm$^{-1}$:

$$CI = \frac{\overline{I}_1([788.25, 796.25 \text{ cm}^{-1}])}{\overline{I}_2([832.31, 834.37 \text{ cm}^{-1}])}, \qquad (3)$$

where $\overline{I}_1$ and $\overline{I}_2$ are the mean radiances of each window.

The AI is defined as the ratio between the mean radiances around the 792 cm$^{-1}$ CO$_2$ band and the atmospheric

window region between 960 and 961 cm$^{-1}$:

$$AI = \frac{\overline{I}_1([788.25, 796.25 \text{ cm}^{-1}])}{\overline{I}_3([960.00, 961.00 \text{ cm}^{-1}])}, \qquad (4)$$

where $\overline{I}_1$ and $\overline{I}_3$ are the mean radiance of each window.

Both the CI and the ACI are continuous values. Low values indicate cloudy air and high values indicate clear air. For the CI, Sembhi et al. (2012) defined a set of variable (latitude, altitude, and season) thresholds to discriminate between clear and cloudy air. For the ACI, a fixed threshold value of 7 provides comparable results

to the CI threshold (Griessbach et al. 2016).

In the second step, a brightness temperature correction method is used to separate aerosol from ice clouds. The resulting aerosol product may contain any type of aerosol, e.g. volcanic ash, sulfate aerosol, mineral dust as well as non-ice polar stratospheric clouds (PSCs).

To study the equatorward transport of the Sarychev aerosol (Sections 4.1 and 4.2), we only use MIPAS data

with ACI < 7 that can cover infrared extinction coefficients as small as $1 \times 10^{-4}$ km$^{-1}$, which corresponds to

$3 \times 10^{-4}$ km$^{-1}$ in the visible wavelength range (Griessbach et al. 2016). For studying the upward transport in the tropics (Section 4.3), we consider all MIPAS retrievals to allow for smaller extinction coefficients.

There are semi-annual data oscillations in the aerosol retrievals. This periodic pattern is caused by the AI that uses the atmospheric window region between 960 and 961 cm$^{-1}$. Around this window region, there are $CO_2$ laser bands. Due to the semiannual temperature changes at about 50 km (semiannual oscillation), the $CO_2$ radiance contribution to this window region also oscillates. As this window is generally very clear of other trace gases, this oscillation is not only visible at higher altitudes but also at altitudes below the temperature changes, because the satellite line-of-sight looks through the whole layer.

**2.3 MPTRAC**

The Massive-Parallel Trajectory Calculations (MPTRAC) model is a Lagrangian particle dispersion model, which is particularly suited to study volcanic eruption events (Heng et al., 2015; Hoffmann et al., 2016). In the MPTRAC model, trajectories for individual air parcels are calculated based on numerical integration of the kinematic equation of motion and simulations are driven by wind fields from global meteorological reanalyses. In this study, the MPTRAC model is driven with the ERA–Interim data (Dee et al., 2011) interpolated on a $1°\times 1°$ horizontal grid on 60 model levels with the vertical range extending from the surface to 0.1 hPa. The ERA–Interim data are provided at 00, 06, 12, and 18 UTC. Outputs of the model simulations are given at the same time interval.

Turbulent diffusion is modelled by uncorrelated Gaussian random displacements of the air parcels with zero mean and standard deviations $\sigma_x = \sqrt{D_x \Delta t}$ (horizontally) and $\sigma_z = \sqrt{D_z \Delta t}$ (vertically). $D_x$ and $D_z$ are the horizontal and vertical diffusivities respectively, and $\Delta t$ is the time step for the trajectory calculations. For the Sarychev simulation, $D_x$ and $D_z$ is assigned to be 50 m$^2$s$^{-1}$ and 0 m$^2$s$^{-1}$ in the troposphere and 0 m$^2$s$^{-1}$ and 0.1 m$^2$ s$^{-1}$ in the stratosphere. Furthermore, the sub-grid scale wind fluctuations are simulated by a Markov model (Stohl et al., 2005; Hoffmann et al., 2016). Loss processes of chemical species, $SO_2$ in our simulations, are simulated based on an exponential decay of the $SO_2$ mass assigned to each air parcel, with a constant half lifetime of seven days.

**3 Simulations and observations of the Sarychev plume**

**3.1 Reconstruction of the Sarychev $SO_2$ emission time series**

The Sarychev peak with summit at 1496 m, is located at 48.1°N, 153.2°E, and it is one of the most active volcanoes of the Kuril Islands. It erupted most recently in June 2009 (VEI = 4). On 11 June 2009, two weak ash eruptions were first detected (Levin et al., 2010) and during the main explosive phase from 12 to 16 June 2009, ash, water vapor, and an estimated $1.2\pm0.2$ Tg of $SO_2$ were injected into the UTLS, making it one of the 10 largest stratospheric injections in the last 50 years (Haywood et al., 2010). Sulfate aerosol was detected several days after the eruption and the enhancement of the optical depth caused by the Sarychev eruption lasted for months, returning to pre-Sarychev eruption values in the beginning of 2010 (Doeringer et al., 2012; Jégou et al., 2013). As shown in Fig. 1 (top), Sarychev is located at the northern edge of the subtropical jet and to the northeast of the ASM (marked by the black rectangle). In the vertical section (Fig. 1, bottom), the dynamical

tropopause, defined by a potential vorticity (PV) value of 2 PV units (PVU, 1 PVU=$10^{-6}$ Km$^2$s$^{-1}$kg$^{-1}$), is around 11 km at the location of the Sarychev.

To reconstruct the altitude-resolved SO$_2$ emission time series, we follow the approach of Hoffmann et al. (2016) and use backward trajectories and AIRS SO$_2$ measurements. Since AIRS measurements do not provide altitude information, we establish a column of air parcels at each location of an AIRS SO$_2$ detection. The vertical range of the column is 0–25 km, covering the possible vertical dispersion range of the SO$_2$ plume in the first few days. The AIRS footprint size varies between 14 and 41 km, so in the horizontal direction we choose an average of 30 km as the full width at half maximum (FWHM) for the Gaussian scatter of the air parcels. In our simulation, a fixed number of 100,000 air parcels is assigned to all air columns and the number of air parcels in each column is linearly proportional to the SO$_2$ index. Then backward trajectories were calculated for all air parcels, and trajectories that are at least 2 days but no more than 5 days long and that have passed the volcano domain are recorded as emissions of Sarychev. The volcano domain is defined to be within a radius of 75 km to the location of Sarychev and 0–20 km in the vertical direction, covering all possible injection heights. Sensitivity experiments have been conducted to optimize these pre-assigned parameters to obtain best simulation results. Haywood et al. (2010) estimated that 1.2 Tg of SO$_2$ were injected into the UTLS on 15 and 16 June 2009 with a 15 % error estimate ($\pm$0.2 Tg). Considering that there were minor emissions before 15 June (Levin et al., 2010; Jégou et al., 2013; Carboni et al., 2016), we allocated a mass of 1.4 Tg to the derived SO$_2$ emissions. The reconstructed SO$_2$ emission time series is shown in Fig. 2.

In Fig. 2, black dots denote the thermal tropopause derived from ERA–interim data at the Sarychev. Sarychev released SO$_2$ almost without interruption in the first five days, but with large variations in height and magnitude. Smaller eruptions began on 12 June followed by continuous eruptions on 13 June, ranging from 7 km to about 17 km. Significant SO$_2$ injections occurred on 14–15 June between 10 and 18 km, followed by minor emissions until 16 June. The majority of SO$_2$ (58%, ~0.81Tg) was injected directly into the extratropical lower stratosphere, and the largest SO$_2$ injection occurred between 12 and 17 km. This time line of the eruptions is consistent with the observation of the Japanese Meteorological Agency Multifunctional Transport Satellite (MTSAT) (Levin et al., 2010; Rybin et al., 2011) and the Optical Spectrograph and Infrared Imaging System (OSIRIS) measurements, showing that the peak backscatter of aerosols was between 12 and 16 km (Kravitz et al., 2011).

The derived SO$_2$ emission time series are the basis of the simulations of SO$_2$ plume in the following sections.

### 3.2 Simulation and validation of the Sarychev plume dispersion

A new set of 100,000 air parcels is assigned to the derived SO$_2$ emission shown in Fig. 2, with 14,000 kg of SO$_2$ in each of the air parcels. The trajectories initialized with this SO$_2$ emission time series are calculated with the MPTRAC model from 12 June 2009 (first eruption) to 31 July 2009. The simulated evolution of the SO$_2$ plume is shown in Fig. 3 (left column) and compared with the AIRS SO$_2$ measurements (right column). Only SO$_2$ column density values larger than 2 Dobson units (DU) are shown. The evolution of simulated plume altitudes is shown in Fig. 4 together with tangent altitudes of the MIPAS aerosol detections. The simulation outputs are given every six hours, but only results at selected time and dates are shown. Figure 3 and Fig. 4 show that, as the SO$_2$ was injected into different altitudes, the SO$_2$ plume split roughly into two branches after the eruption, moving eastwards and westwards, and at the same time, most of the emissions moved poleward. On 22 June, the

SO$_2$ plume over Eastern Siberia stretched towards three directions: northeast, south, and south east. The SO$_2$ in the elongated filament over the Eastern Siberia and North-east China with altitudes below 9 km was diluted and partly depleted and converted into aerosol afterwards and the other two filaments moved toward east. The SO$_2$ plume over the Northwest American continent stretched towards east and west, forming a long filament running through Northern Canada. The SO$_2$ concentration declined exponentially and only a fraction of SO$_2$ remained near the Bering Strait and Northern Canada till 28 June.

In Fig. 3, in order to validate the simulation of plume dispersion, and also to indirectly validate the reconstructed emission time series, we compare the simulations of plume dispersion with AIRS SO$_2$ measurements. The SO$_2$ index from AIRS detections was converted into SO$_2$ column density using the correlation function described in Hoffmann et al. (2014), which was built using radiative transfer calculations. AIRS was able to detect the SO$_2$ cloud from the beginning of the eruption of Sarychev up to five weeks later (not fully shown in the figure). The SO$_2$ column density derived from AIRS agrees well with the SO$_2$ column density derived from the Infrared Atmospheric Sounding Interferometer (IASI) in magnitude, e.g. see Fig. 3 in the study of Haywood et al. (2010) and Fig. 2 in the study of Jégou et al. (2013). Generally, the simulations agree well with the AIRS measurements in position and diffusion and the simulations can provide more information on the SO$_2$ distribution than the AIRS measurements alone. The differences between the SO$_2$ clouds, e.g. on 24 June (over the western Pacific) are partly attributed to a mismatch in time of the AIRS SO$_2$ measurements and the simulation output. In magnitude, the SO$_2$ column density from AIRS is slightly larger than that from MPTRAC simulations, and the SO$_2$ maxima are found in different locations. This is also found by Hoffmann et al. (2016) for other eruption events and this was attributed to the fact that the inverse modelling approach is optimized to reproduce the spatial extent of the plume but not the absolute emission. Except for the discrepancy between the times of the compared data, this remaining difference may also be attributed to the initial setting of the total SO$_2$ mass, the SO$_2$ life time and the uncertainties of the ERA-interim winds.

In Fig. 4, the altitudes of the simulated volcanic plume are compared with the tangent altitudes of MIPAS aerosol detections to verify the vertical distribution of the air parcels from the model. In the beginning of the simulation, the volcanic plume was composed of SO$_2$ but the SO2 was converted to sulfate aerosol during the transport process. Here we assume that the SO$_2$ remains collocated with the sulfate aerosol and the sedimentation of the small sulfate aerosol particles is negligible for the time scale considered. Aerosol produced from the SO$_2$ emissions as detected by MIPAS within a few days after the initial eruption. In general, the altitudes of the SO$_2$ air parcels from the MPTAC model are comparable to the MIPAS aerosol altitudes. The majority of the air parcels were between 10 and 20 km and moved eastwards. A thin filament over the north Pacific ascended to up to 20 km and moved westward to East Asia by the end of June 2009, which was well verified by the MIPAS detections. Some inconsistencies appeared, e.g., along the west coast of North America on 23 June and over the northeast Pacific on 24 and 25 June. We attributed them to the fact that the SO$_2$ at lower altitudes (below 14 km) had been converted into aerosol more quickly than the SO$_2$ at higher altitudes (above 16 km). The various vertical distributions of the air parcels were also verified by some overlapping MIPAS detections, e.g., over northwest America. Overall, this comparison demonstrates that the MPTRAC simulation provides a quite accurate representation of the observed horizontal and vertical distribution of the Sarychev SO$_2$ and aerosol plume.

## 4 Equatorward dispersion of the Sarychev plume and the role of the ASM

### 4.1 Equatorward dispersion of Sarychev plume

Although the majority of the Sarychev plume was transported towards the north pole, our simulations show that there was a clear equatorward dispersion from the extratropical stratosphere to the tropical lower stratosphere (Fig. 5). The top panel of Fig. 5 shows the time and latitude distribution of the air parcels above the height of the 380 K isentropic surface in percentage of the total number of air parcels in each bin. The bin size used here is 12 hours $\times 2\,^\circ$ in latitude. The plume reached the tropical stratosphere about one week after the eruption, and crossed the Equator by the end of June 2009. The red dots in the top panel of Fig. 5 denote the air parcels between 30 $^\circ$N and 30 $^\circ$S above the 380 K isentropic surface in proportion to the total air parcels released in the forward trajectory simulation. Until the end of July 2009, nearly 4% of the air parcels entered the tropical stratosphere.

This finding is verified by the MIPAS aerosol detections (Fig. 5). The middle panel of Fig. 5 shows the aerosol above the 380 K isentropic surface. The altitudes between 380 and 400 K isentropic surfaces are marked with diamonds and detections above the 400 K isentropic surface are marked with circles with a different color scheme. The bottom panel of Fig. 5 shows that shortly after the eruption of Sarychev, large quantities of stratospheric aerosol formed from directly injected $SO_2$. On 31 July 2009, the end of the MPTRAC trajectory simulation, the MIPAS aerosol detections between 30 $^\circ$N and 30 $^\circ$S above the 380 K isentropic surface were increased by about six times compared with the number before the Sarychev eruption. And the increase of tropical stratospheric aerosol continued until early September 2009. The number of aerosol detections between 30 $^\circ$N and 30 $^\circ$S above the 400 K isentropic surface also increased but the peak appeared around end of September 2009, later than the peak of the aerosol above the 380 K isentropic surface.

The aerosol in the extratropical stratosphere was removed by the end of 2009, while the aerosol that had entered the tropical stratosphere stayed for months. Aerosol optical depths from OSIRIS and scattering ratio from CALIPSO lidar measurements have also shown similar equatorward dispersion of the aerosol after the eruption of Sarychev (Haywood et al., 2010; Solomon et al., 2011).

### 4.2 The role of the ASM anticyclone

As shown by the evolution of Sarychev plume in section 3.2, active Rossby wave breaking events at mid-latitudes during boreal summer have significantly influenced the plume dispersion. In boreal summer, the ASM is among the most prominent circulation patterns. Figure 6 shows a cross section of the ASM in boreal summer 2009. Generally, the ASM anticyclone ranged from 360–400 K, marked by the negative PV anomaly of -3 PVU and it was bounded by the subtropical and equatorial jets. The thermal tropopause averaged over 40–120 $^\circ$E was elevated up to about 390 K. The ASM anticyclonic circulation facilitates meridional transport when the subtropical jet weakens and retreats northward (Haynes and Shuckburgh, 2000). Figure 7 depicts the MIPAS aerosol measurements between 40–120 $^\circ$E during June–August 2009. The aerosol detections between 360 and 400 K to the south of 40 $^\circ$N were collocated with the core of the ASM, extending from the mid-latitude UTLS to the TTL. The aerosol that made its way above the subtropical jet core to lower latitudes along the isentropic surfaces cannot be explained by convection but only by large-scale transport associated with the ASM.

Using our trajectory simulations, it is straightforward to see the role of ASM anticyclonic circulation in influencing the pathway of the plume. The distributions of air parcels in the vertical range of 360–400 K (in

percentage of total number of air parcels released after the eruption) on 30 June, 10 July, 20 July and 31 July 2009 are shown in Fig. 8. The red contour is the geopotential height of 14,320 m on the 150 hPa pressure surface, denoting a commonly used boundary of the ASM anticyclone (Randel and Park, 2006). At latitudes between 15–40 °N in summer, the 150 hPa pressure surface is around 370 K .The black contour is the PV-based ASM  transport barrier for July 2009 (1.8 PVU) on 370 K isentropic surface as defined by Ploeger et al. (2015). In the first 19 days (top left panel in Fig. 8), from first eruption on 12 June to 30 June 2009, the plume mostly moved eastward and remained at mid- and high latitudes. After another 10 days (top right panel in Fig. 8), air parcels were entrained into the anticyclonic circulations of the American monsoon and a fraction was shed towards the tropics. The remaining air parcels that entered the ASM circulation were entrained along the flow surrounding the ASM anticyclone and moved south-westward approaching the tropics. In the following 20 days (bottom panels in Fig. 8), some air parcels were dragged along the flow south of the ASM, and some were shed out from the south-eastern flank of the anticyclone and spread over the tropics. The American monsoon plays a similar role as the ASM in transporting air parcels to lower latitudes, but it is much weaker in strength. The 'aerosol hole' between 360 and 400 K illustrated the ASM's role as a transport barrier between the air inside and outside of the anticyclone. In our case, it is better demarcated by the PV-based barrier than the geopotential height criterion. The northern barrier of the subtropical jet was strong, while the southern barrier was more permeable for meridional transport.

This meridional transport under the influence of the ASM revealed by the simulations shown in Fig. 8 is verified by the MIPAS aerosol detections shown in Fig. 9. Because of the sparse horizontal coverage of MIPAS detections, 1.5-day forward and 1.5-day backward trajectories initialized by MIPAS aerosol detections were calculated to fill gaps in space and time. The trajectories are initialized with MIPAS aerosol detections from 12 June 2009 to 31 July 2009. These detections are traced both forward and backward for 1.5 days with the MPTRAC model and the model output is provided every six hours. So this calculation produces 12 times as many aerosol data points as the original MIPAS detections. Compared with MIPAS detections in Fig. 9, the simulations in Fig. 8 could successfully reproduce the maxima, minima and filaments of the aerosol distributions. It is also verified by MIPAS detections that the transport barrier of ASM is better demarcated by the PV-based barrier.

The transport pathway to low latitudes for aerosols above the 400 K isentropic surface is shown by simulations in Fig. 10 and MIPAS aerosol detections in Fig. 11. At altitudes above the ASM, trajectories are driven by easterlies and meridional, isentropic winds. Until the end of July 2009, air parcels above the 400 K isentropic surface were transported to lower latitudes (as far as about 15 °N), but could not reach the Equator. This suggests that ASM play the most significant role between the 360 and 400 K isentropic surfaces, which may vary in spatial extent associated with the strength of the ASM.

In addition, a sensitivity test simulation of the same eruption in winter (January 2009) shows that, in northern hemisphere winter, meridional transport from the extra-tropic to the tropics is typically suppressed by the strong winter subtropical jets.  Forward trajectories initialized by the Sarychev $SO_2$ emissions (as shown in Fig. 2), but driven by winter wind fields in January and February 2009 are shown in Fig. 12. Compared with the summer scenario in Fig. 8, wind speeds in winter 2009 between 360 and 400 K were faster, and the trajectories could span the northern hemisphere within 20 days. About 40 days after the eruption, air parcels were almost evenly distributed at mid- and high latitudes, but no air parcels did approach the Equator.

### 4.3 Upward transport of Sarychev aerosol

Simulation results from Figs. 8 to 11 have shown that the ASM anticyclone enhanced the equatorward dispersion of the Sarychev aerosol in the vertical range of 360–400 K, but the ASM did not facilitate the equatorward dispersion above 400K. The increased aerosol in the tropical stratosphere above 400 K (~ 18 km) as seen in the middle and bottom panels of Fig. 5 could only be explained by the upward transport above the TTL. Above the zero diabatic heating surface (generally around the 360 K isentropic surface), air masses that

enter the TTL are considered to be lifted effectively by radiative heating (Gettelman et al., 2004; Fueglistaler et al., 2009).

In Fig. 13, the upward transport of aerosol in the tropics is clearly demonstrated by the MIPAS ACI at high altitudes between 10 °N and 10 °S. After the eruption of Sarychev 2009, at 15 km, an enhanced aerosol signal was found from July 2009 and lasted for about 10 months. It returned to pre-Sarychev level at the end of May

2010. At 20 km, an enhanced aerosol signal was detected in October 2009, and in another four months, the enhanced aerosol signal was found at 25 km. It should be noted that the green shades in early 2009 and the dark red shades above the green shades at about 20-26 km in late 2009 show aerosol from the Kasatochi eruption. The green shades in late 2010 and early 2011 are aerosol from the Merapi eruption in October-November 2010. The overlaid speed of the tropical (10 °N–10 °S) tape recorder signal, namely the slope of water vapor isolines in

a time–height plot, is about 10 hPa per month. It is a measure of the effective upward speed of the BDC and it is shown from the altitude of the base of the upward BDC in the tropics (~70 hPa). The aerosol transported to the TTL is moved upward by the BDC. The ascent speed of the Sarychev aerosol agrees well with the ascent speed of the water vapor tape recorder before and after the data oscillation at the end of 2009 and beginning of 2010. The "aerosol tape recorder" shown by MIPAS data here has first been reported by Vernier et al (2009) in their

Section 2.4 using Cloud-Aerosol Lidar with Orthogonal Polarization (CALIOP) data. Other satellite sources, e.g., Optical Spectrograph and InfraRed Imager System (OSIRIS) time series, also have shown this upward transport (Rieger et al. 2015), but the upward transport were not discussed in these publications.

### 5 Discussion

The results of our study in the above sections suggest that the ASM anticyclone plays a key role in transporting

the Sarychev aerosol from the extra-tropical lower stratosphere into the TTL. In various studies, this quasi-isentropic transport from extra-tropics to the TTL is referred as in-mixing process (e.g., Konopka et al., 2009). Horizontal in-mixing of tracer species has been observed (e.g. Folkins et al., 1999) or modelled (e.g. Konopka et al., 2010), and has been used to explain the seasonal and annual cycle of tracer species (Abalos et al., 2013; Ploeger et al., 2013). The role of in-mixing is prominent when there are large gradients in these tracers between

the tropics and the extra-tropics. The study of Abalos et al. (2013) shows that the main contribution to in-mixing originates in the northern hemisphere and is related to the Asian monsoon, and this in-mixing process takes place in the TTL close to the tropopause. These extra-tropics to tropics transport events are also considered to be driven by anticyclonic Rossby wave breaking (Homeyer and Bowman, 2013). The net equatorward transport peaks downstream of large anticyclones in the potential temperature range between 370 and 390 K (Homeyer

and Bowman, 2013). Above the TTL, in-mixing rapidly decreases with height and becomes very weak at altitudes of the tropical pipe (Ploeger et al., 2013). These findings agree well with our study, which shows that

the ASM anticyclonic circulation enhanced the equatorward transport between 360 and 400 K, but not above 400 K.

The pathway of the Sarychev plume approaching the deep tropics is modulated by the transport barrier at the boundary of the ASM anticyclone as well. The 'aerosol hole' shown in Figs. 8 and 9 that ranges from 360 to 400 K and collocates with the core of ASM anticyclone, only appears during the ASM season. Conventionally, the geopotential height of 14,320 m on the 150 hPa pressure level is used to define the boundary of the ASM anticyclone, but from the perspective of transport, the PV-based barrier defined by Ploeger et al. (2015) can better represent the boundary.

Our results show that the meteorological background conditions during a volcanic eruption have a significant impact on the transport of the volcanic aerosol. For instance, the Puyehue-Cordón Caulle emissions reached the lower stratosphere and were rapidly transported eastward by the jet stream (Klüser et al, 2013; Hoffmann et al., 2016), while the Nabro emissions were entrained first by the ASM circulation in UTLS region before being transported to the tropics (Fairlie et al., 2014; Heng et al., 2016). In this study, the transport of the equatorward dispersion of Sarychev aerosol is driven by the ASM anticyclone. Aerosol entering the TTL via the ASM anticyclone further entered the ascending branch of the BDC. This enabled the Sarychev aerosol to remain in the stratosphere for months and further spread over both hemispheres. In this way, the Sarychev eruption may not only influence the northern hemisphere, but could also have potential impact on the global chemical composition and radiative budget similar to a tropical volcanic eruption.

Based on our model simulation, 4% of the air parcels that we released as the $SO_2$ injected into the atmosphere by the Sarychev eruption has been transported in between 30 °N and 30 °S above the 380K isentropic surface. This result could be affected by many possible mechanisms of aerosol loss, like the interaction of sulfate aerosol and clouds, and coagulation in the volcanic plume or with other particles. Larger particle size may result in quicker sedimentation rates, especially at higher altitudes where the mean free path between air molecules far exceeds the particle size and particles fall more rapidly than they would otherwise. The scavenging efficiency of $SO_2$ could be increased if it is incorporated into growing ice particles (Textor et al., 2003). Also, $SO_2$ is slightly soluble in liquid water and it may have a small chance to be washed out during the transport process. But as revealed in our study, the efficient pathway of the transport is approximately between the 360 and 400 K isentropic surfaces, where the atmosphere is relatively dryer, cooler and cleaner than in the lower troposphere. So our model result can be considered as a representative value. Although only a fraction of the $SO_2$ (~4% out of 1.2±0.2 Tg) was transported to the tropical stratosphere by the end of July 2009, if the $SO_2$ is entirely converted into gaseous $H_2SO_4$ and condensed into a 75%-25% $H_2SO_4$-$H_2O$ solution, the total aerosol mass loading added to the tropical stratosphere by the Sarychev eruption would be about 0.06±0.01 Tg of sulfate aerosol, which is several times larger than the 0.015–0.02 Tg per year required to explain the background aerosol increase of 4–7 % per year after 2002 as found by Hofmann et al. (2009) at the Mauna Loa Observatory in Hawaii (19.5 °N). Moreover, comparing with the time period from 12 June 2009 to 31 July 2009, larger amounts of aerosols were transported to or ascended to the tropical stratosphere after July 2009.

Since the potential climate impact of high-latitude volcanic emissions largely depends on their plume height and the meteorological background conditions, it is essential to conduct Lagrangian transport simulations initialized with realistic time- and altitude-resolved $SO_2$ emission time series. The backward trajectory approach used in

this study to reconstruct the emission time series proves to be an efficient way to produce realistic $SO_2$ emission time series.

## 6 Summary

In this study, we analyzed the equatorward transport pathway of volcanic aerosol from the high-latitude volcanic eruption of the Sarychev in 2009. The analysis was based on MIPAS aerosol detections, AIRS $SO_2$ measurements and trajectory simulations.

First, the time- and altitude-resolved $SO_2$ injection was derived using backward trajectories initialized with AIRS $SO_2$ measurements. Second, the dispersion of Sarychev plume from the beginning of the eruption (12 June 2009) to 31 July 2009 was simulated based on the derived $SO_2$ emission time series. The horizontal distribution of the plume and its altitudes were validated with AIRS $SO_2$ measurements and MIPAS aerosol measurements respectively. The comparisons showed that there was good agreement between the simulations and observations.

The results presented in this study suggest that in boreal summer, the transport and dispersion of volcanic emissions were greatly influenced by the dominating ASM circulation, which facilitates the meridional transport of aerosols from the extra-tropical UTLS to the TTL by entraining air along the anticyclonic flows and shedding the air to the deep tropics downstream of the anticyclonic circulation. Meanwhile, the ASM anticyclone effectively isolates the air inside of the ASM from aerosol-rich air outside the anticyclone. The transport barrier at the boundary of the ASM is better denoted by the barrier defined by the PV gradient approach. However, the ASM only influenced the plume dispersion significantly in the vertical range of 360–400 K. The ASM anticyclone did not have notable impact on the equatorward dispersion above 400 K, where the Sarychev aerosol was confined in the northern hemisphere.

The simulations show that until the end of July 2009, 0.06±0.01 Tg of sulfate aerosol had entered the tropical stratosphere. The MIPAS measurements showed a continuous increase of the aerosol load in the tropical stratosphere until early September 2009. After entering the TTL, the aerosol experienced large-scale ascent in the BDC. The ascent speed agrees well with the ascent speed of the water vapor tape recorder. Aerosol signal in the tropics was enhanced within one month of the eruption, and returned to pre-Sarychev level after about 10 months.

The Sarychev eruption had the chance to contribute to the stratospheric aerosol loading in the tropics because it occurred in boreal summer when the equatorward transport was enhanced by the ASM anticyclone. If the eruption had happened in winter, the volcanic aerosol would be confined by the strong subtropical jet to the northern hemisphere.

## 7 Code and data availability

AIRS data are distributed by the NASA Goddard Earth Sciences Data Information and Services Center (AIRS Science Team and Chahine, 2007). The $SO_2$ index data used in this study are available for download at https://datapub.fz-juelich.de/slcs/airs/volcanoes/ (last access: 21 Agu 2017). Envisat MIPAS Level-1B data are distributed by the European Space Agency. The ERA–Interim reanalysis data were obtained from the European Centre for Medium-Range Weather Forecasts. The code of the Massive-Parallel Trajectory Calculations

(MPTRAC) model is available under the terms and conditions of the GNU General Public License, Version 3 from the repository at https://github.com/slcs-jsc/mptrac (last access: 12 April 2017).

## 8 Competing interests

The authors declare that they have no conflict of interest.

*Acknowledgements*. This work was supported by National Natural Science Foundation of China under grant No. 41605023 and International Postdoctoral Exchange Fellowship Program 2015 under grant No. 20151006. The PV gradient based transport barrier data is provided by Dr. Felix Ploeger, Jülich.

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

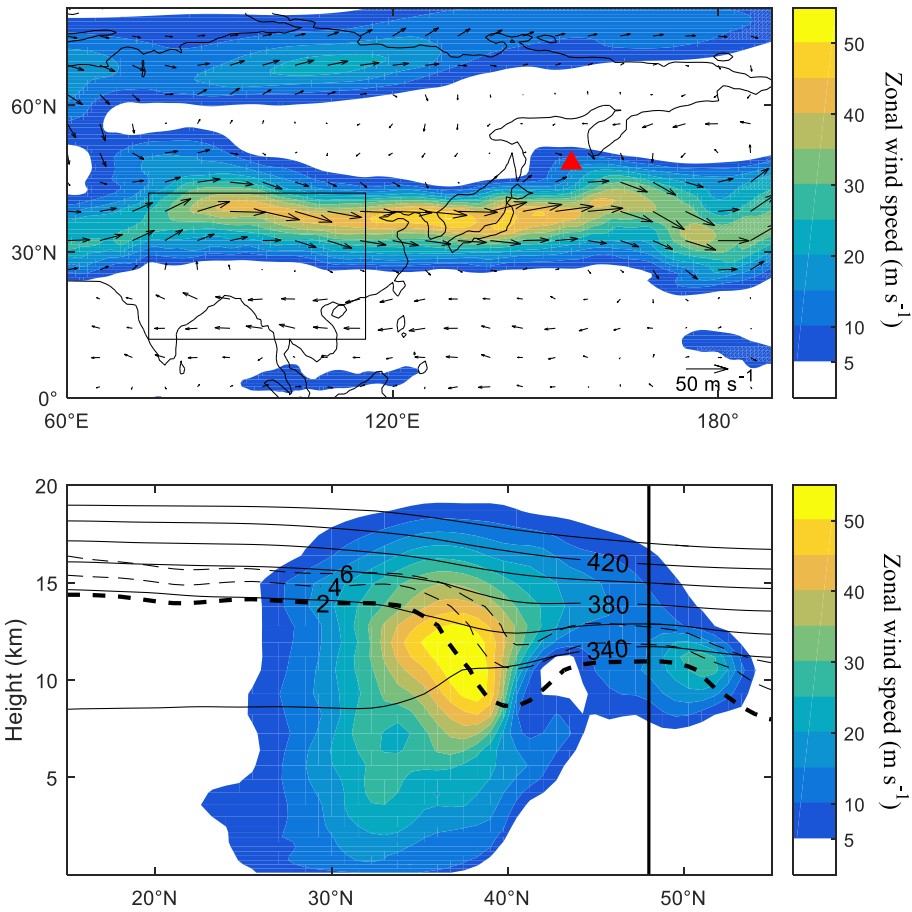

**Figure 1: Meteorological conditions at the time of the Sarychev eruption (based on ERA–Interim reanalysis). Top: zonal wind (shaded) and wind vectors on the 370 K isentropic surface at 0 UTC on 12 June 2009. The location of the Sarychev peak is denoted with a red triangle. The black rectangle indicates the area of the developing ASM anticyclone. Bottom: vertical section of zonal wind (shaded) and contours of potential temperature from 340 K to 440 K (black solid lines) and potential vorticity from 2 PVU to 6 PVU (black dashed lines) along 153°E. The vertical black line denotes the latitude (48.1°N) of the Sarychev peak.**

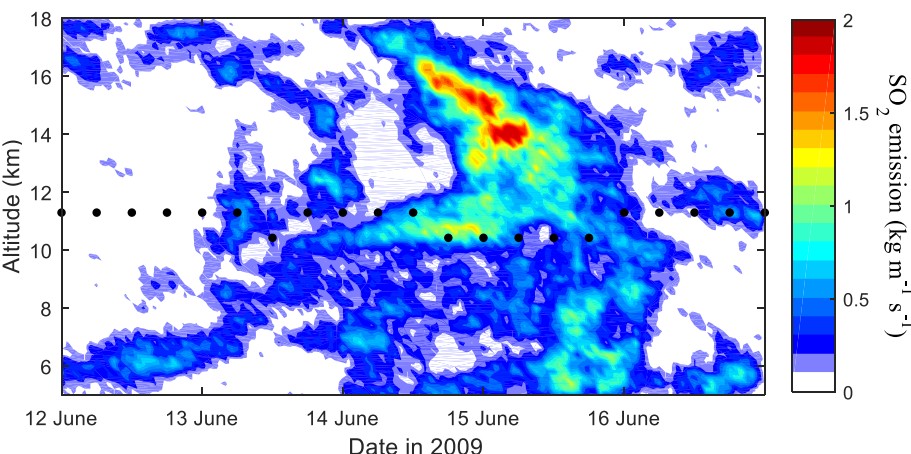

**Figure 2: Sarychev SO₂ emission time series derived with AIRS measurements using a backward trajectory approach (see text for details). The emission data is binned every 1 hour and 0.2 km. Black dots denote the height of the thermal tropopause.**

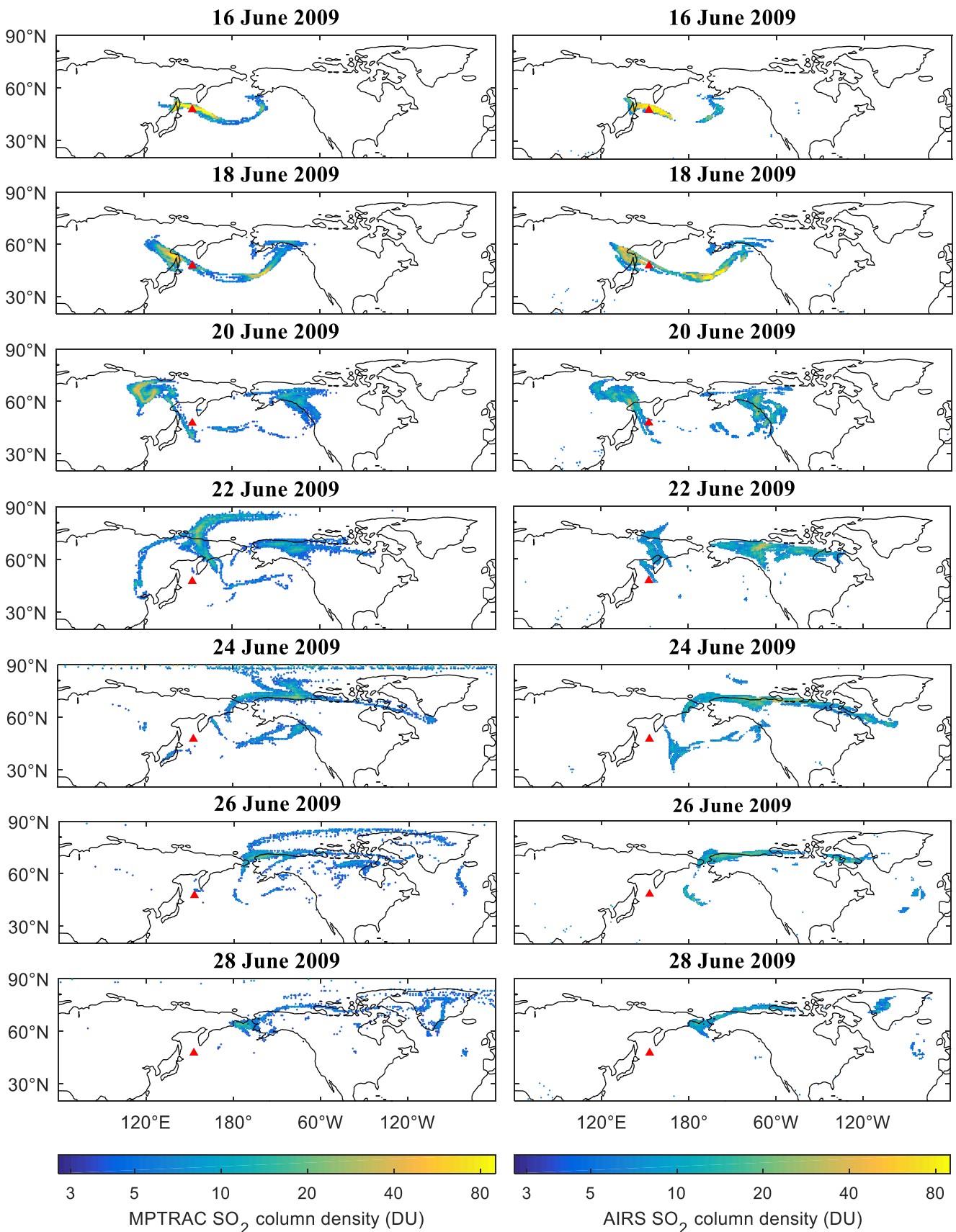

**Figure 3: The evolution of SO$_2$ column density from MPTRAC simulations (left) and AIRS observations (right) for the period 16–28 June 2009. The MPTRAC SO$_2$ column densities are shown for 0 UTC on selected days and AIRS data are collected within ± 6 hours. Only values larger than 2 DU are shown. The red triangle denotes the location of the Sarychev peak.**

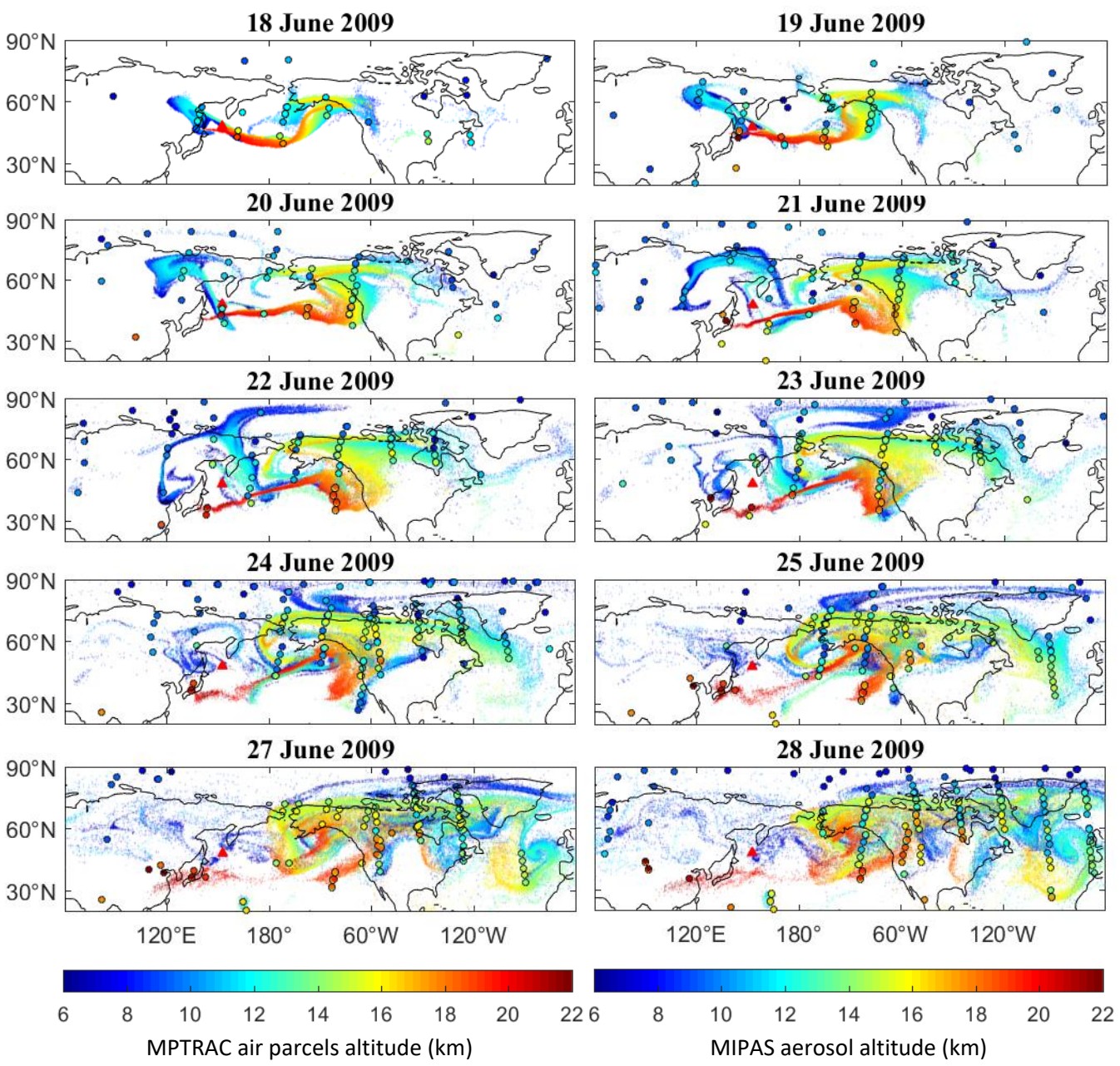

**Figure 4: The evolution of SO₂ air parcel altitudes (shading) from MPTRAC simulations (shown for 0 UTC on selected days) and MIPAS aerosol detections within ± 6 hours (color-filled circles). The altitudes of all air parcels, regardless of their SO₂ values, are shown. The red triangle denotes the location of the Sarychev peak.**

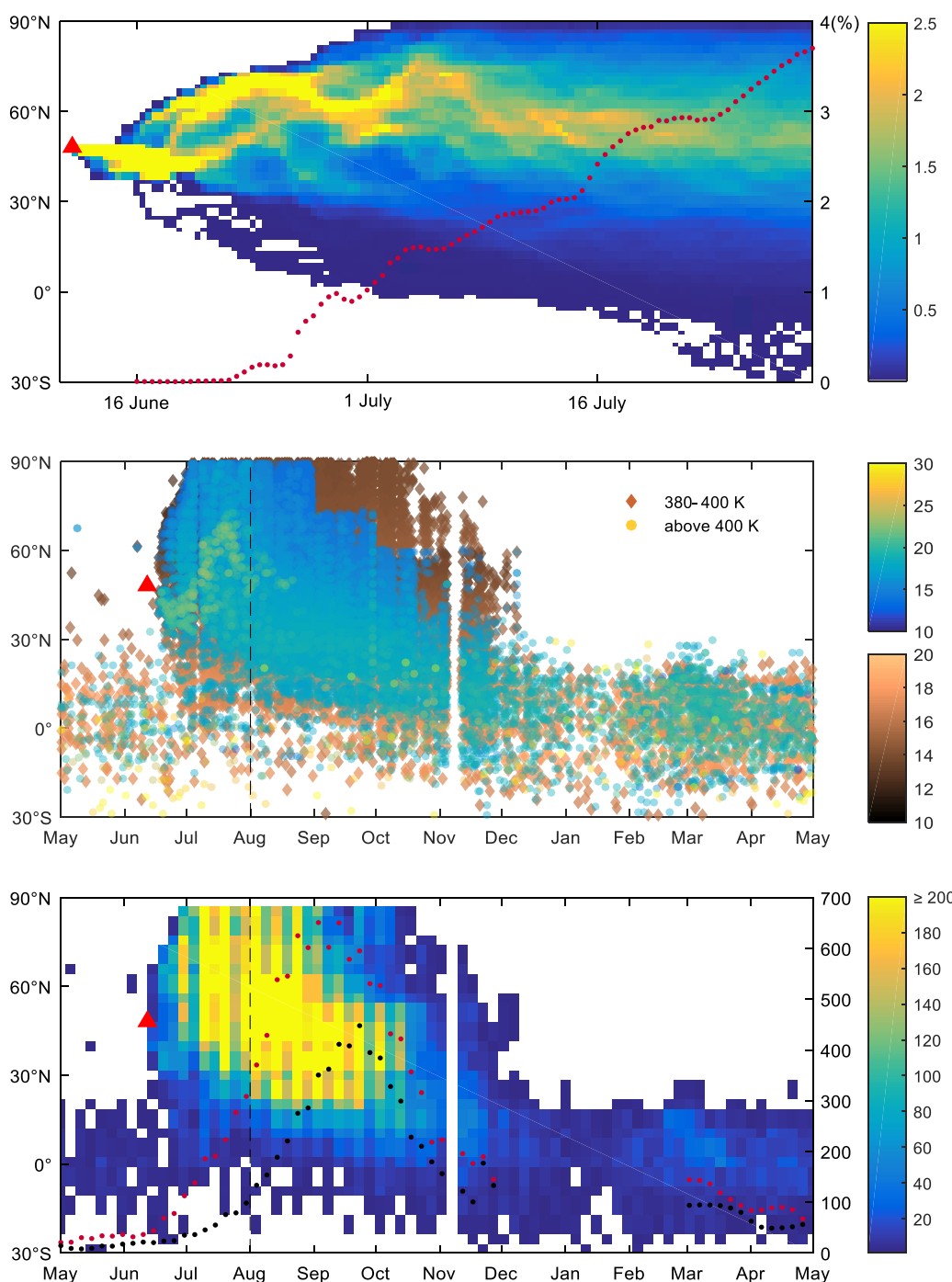

**Figure 5: Top: Percentage (%) of air parcels above the 380 K isentropic surface from MPTRAC simulations, binned every 12 hours and 2 ° in latitude. Red dots denote the percentage of air parcels above the 380 K isentropic surface between 30 ˚N and 30 ˚S. Middle: Altitudes (km) of MIPAS aerosol detections above the 380 K isentropic surface from May 2009 to April 2010. Bottom: Number of MIPAS aerosol detections above the 380 K isentropic surface from May 2009 to April 2010. Detections are binned every 5 days and 5 ° in latitude. Red dots denote the number of the MIPAS aerosol detections above the 380 K isentropic surface between 30 ˚N and 30 ˚S. Black dots denote the number of the MIPAS aerosol detections above the 400 K isentropic surface between 30 ˚N and 30 ˚S. The number during the MIPAS data gap in 2009–2010 winter (Dec, Jan, and Feb) is omitted. The red triangle denotes the time and latitude of the first Sarychev eruption. Dashed lines in the middle and bottom panels indicate the time corresponding to the end of the MPTRAC simulation in the top panel, i.e., 31 July 2009.**

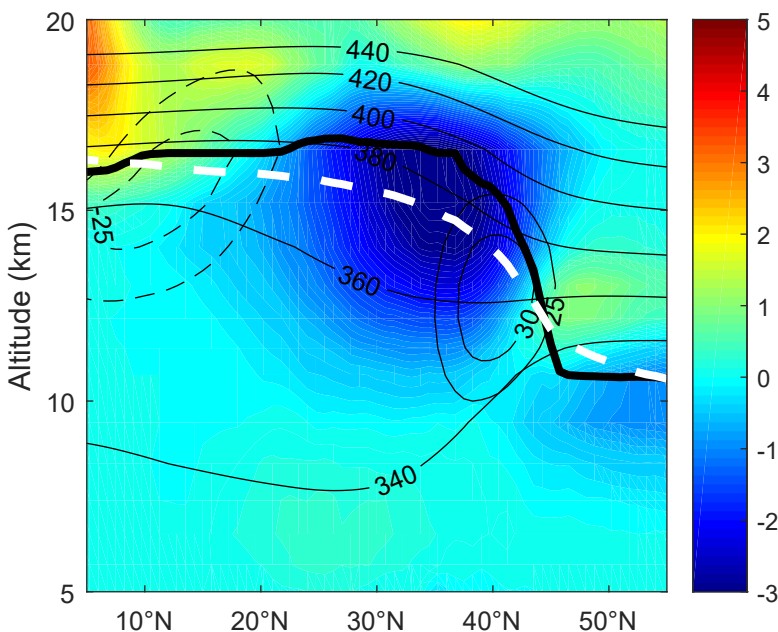

**Figure 6: Meteorological conditions in the Asian monsoon anticyclone (based on ERA–Interim reanalysis). PV anomaly (shaded, unit: PVU) in the Asian monsoon anticyclone (40–120°E) with respect to the zonal mean, averaged over 2009 summer (June-August). Zonal wind (black solid lines indicating the westerlies and black dashed lines indicating the easterlies) is averaged between 40 and 120°E. The first thermal tropopause zonally averaged over 0–360°E is shown as dashed white line, averaged over 40 to 120°E as thick black line.**

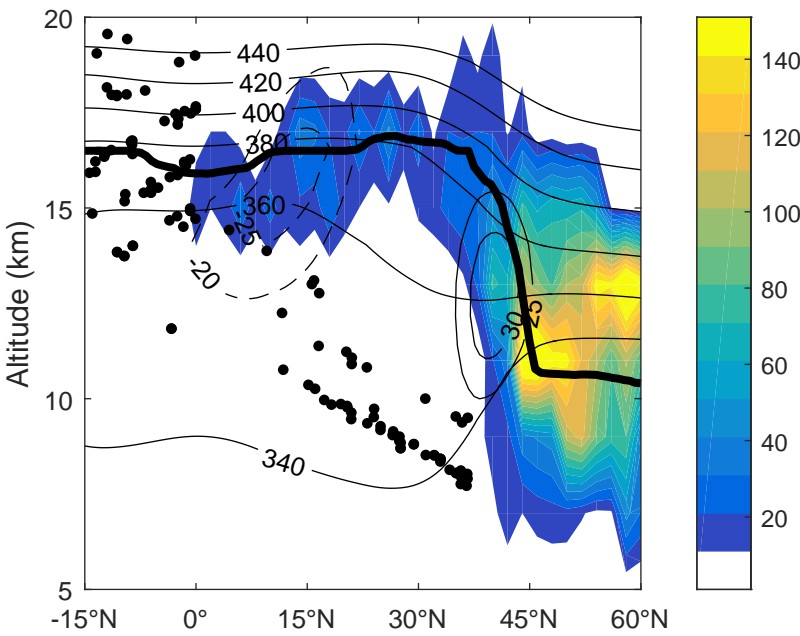

**Figure 7: Number of MIPAS aerosol detections between 40°E and 120°E during June-August 2009 (binned every 2 km in altitude and 2° in latitude). Sparse detections (number of detections in each bin is smaller than 10 are shown with black dots). The tropopause, potential temperature, and zonal wind are the same as shown in Fig. 6.**

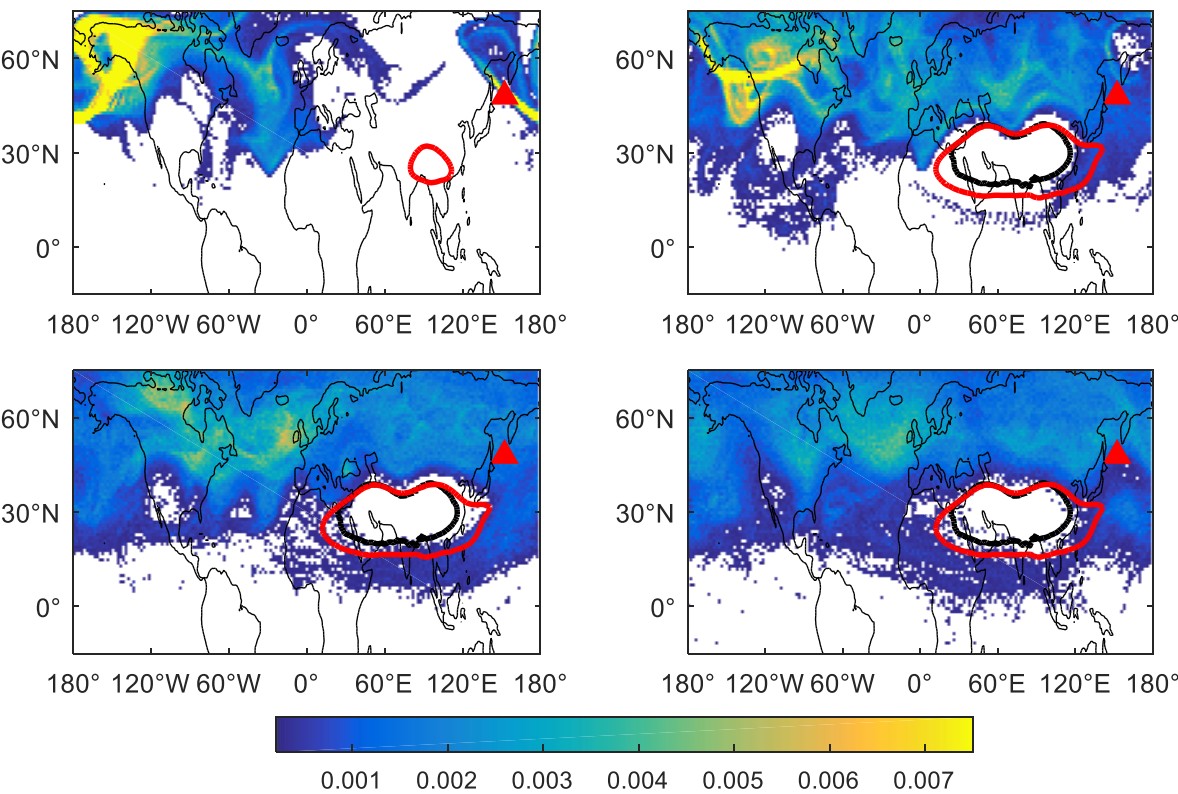

**Figure 8: Percentage (%) of air parcels for 12–30 June (top left), 01–10 July (top right), 11–20 July (bottom left) and 21–31 July 2009 (bottom right) between isentropic surfaces of 360 and 400 K from MPTRAC simulations. Results are binned every 2° in longitude and 1° in latitude. The monthly mean 14,320 m geopotential height on the 150 hPa pressure surface is marked in red and the monthly mean PV-based barrier on the 370 K isentropic surface is marked in black. A PV-based barrier related to the Asian summer monsoon was not identified in June 2009. The red triangle denotes the location of the Sarychev volcano.**

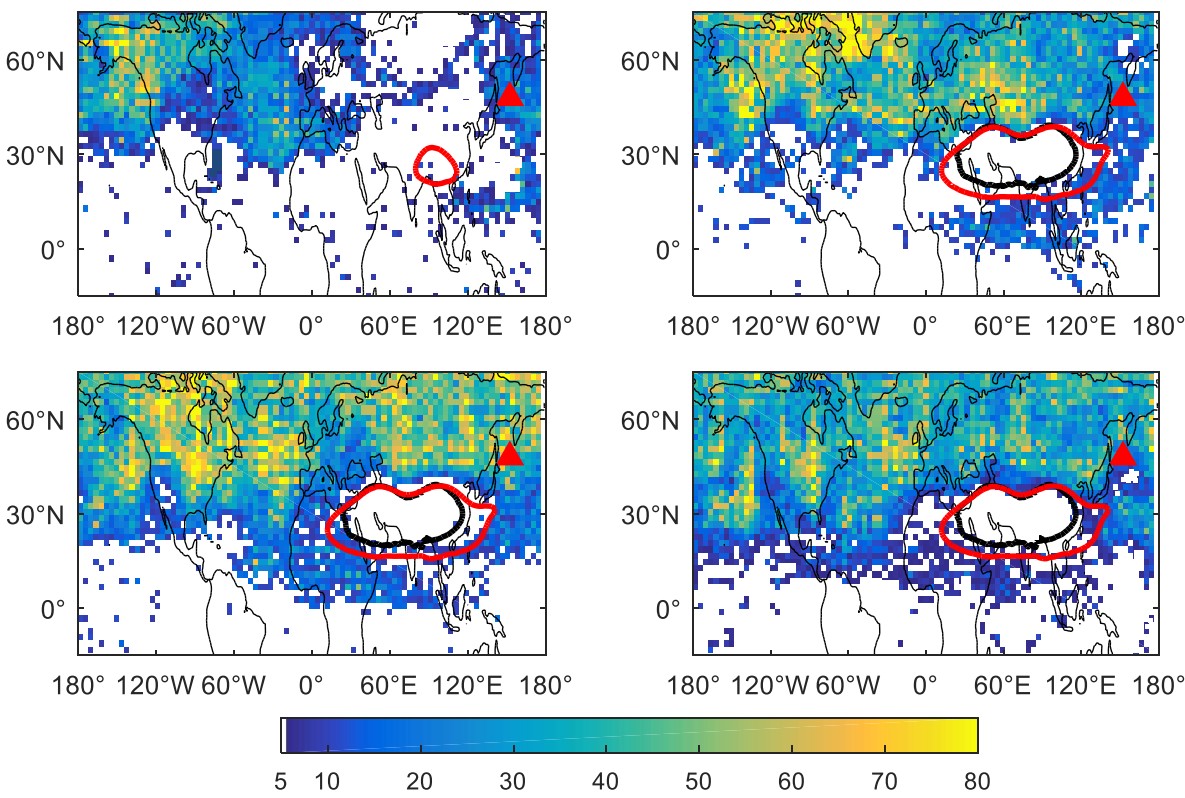

**Figure 9: Number of aerosol data points generated with 1.5-day forward and 1.5-day backward trajectory calculation, for 12–30 June (top left), 01–10 July (top right), 11–20 July (bottom left) and 21–31 July 2009 (bottom right) between isentropic surfaces of 360 and 400 K. Results are binned every 4° in longitude and 2° in latitude. The 14,320 m geopotential height on the 150 hPa pressure surface is marked in red and the PV-based barrier on the 370 K isentropic surface is marked in black. The red triangle denotes the location of the Sarychev volcano.**

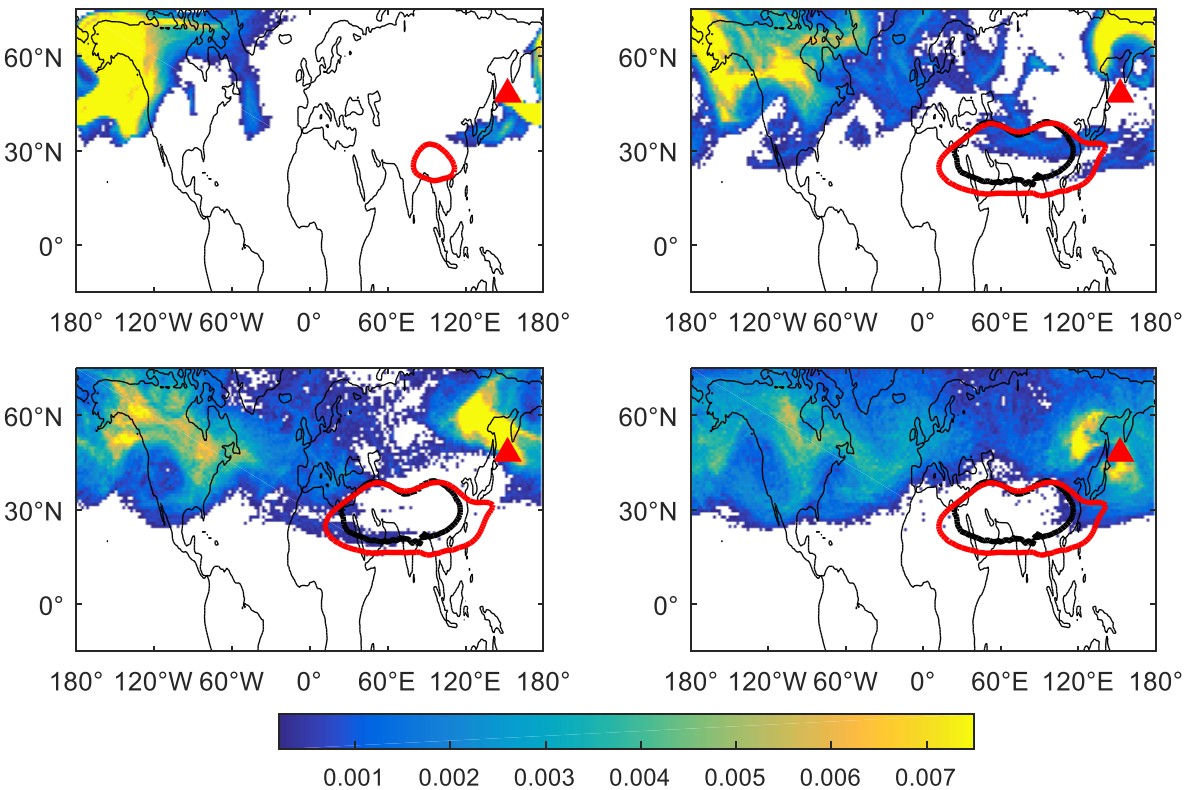

**Figure 10: Same as Fig. 8 but for percentage (%) of total air parcels above the 400 K isentropic surface.**

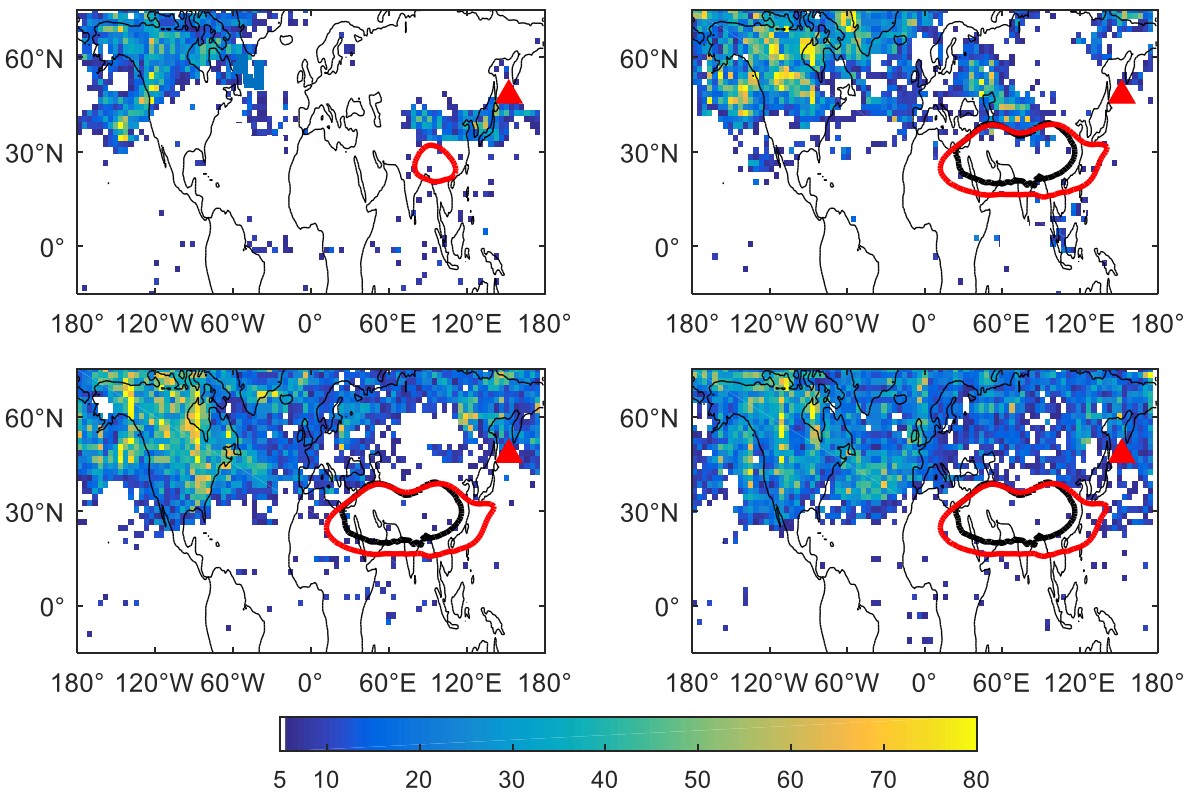

**Figure 11: Same as Fig. 9, but for number of data points above the 400 K isentropic surface.**

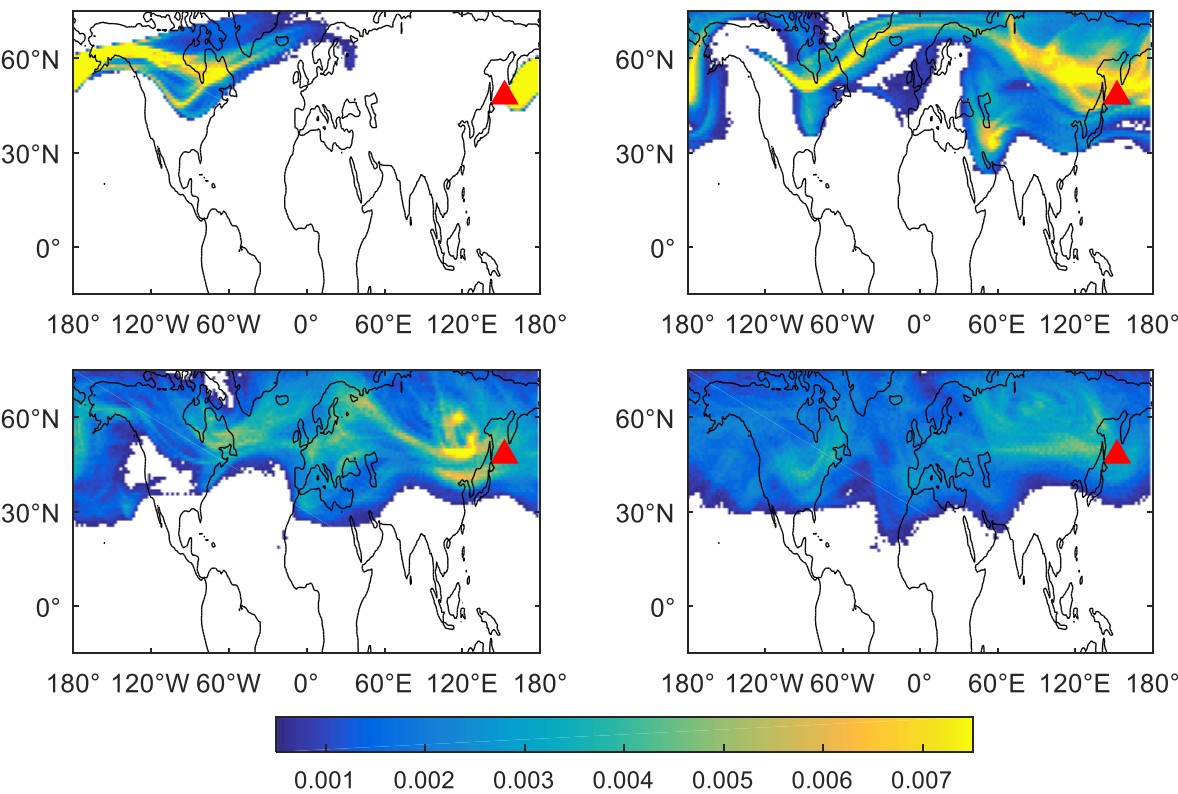

**Figure 12: Percentage (%) of air parcels for the wintertime sensitivity study, on 10 January (top left), 20 January (top right), 31 January (bottom left) and 10 February 2009 (bottom right) from an MPTRAC simulation for a hypothetical eruption of the Sarychev between isentropic surfaces of 360 and 400 K. Results are binned every 2° in longitude and 1° in latitude. The red triangle denotes the location of the Sarychev volcano.**

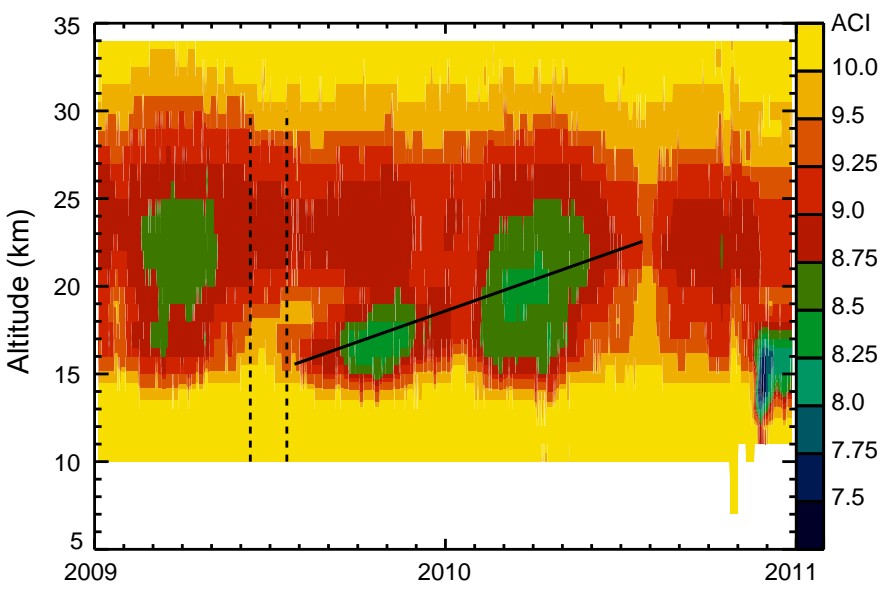

**Figure 13: MIPAS nine-day running median of aerosol-cloud-index (ACI) between 10°N and 10°S (averaged along longitudes) from 2009 to 2010, overlaid with ascent speed of the water vapor tape recorder (solid line) derived with Aura Microwave Limb Sounder (MLS) water vapor mixing ratio from Glanville et al. (2017). Black dashed lines indicate the 12 June 2009 when the eruption started and 20 July 2009 when the simulations show first substantial aerosol transport to 10°N–10°S.**