# Peer review of "Equatorward dispersion of a high-latitude volcanic plume and its relation to the Asian summer monsoon: a case study of the Sarychev eruption in 2009"

_Atmospheric Chemistry and Physics, 2017_

## Referee Comment (RC1) · Anonymous Referee #1 · 16 Jun 2017

Wu et al. present a case study of the dispersion of the volcanic plume from the Mt. Sarachev eruption in 2009. They effectively demonstrate the use of AIRS SO2 observations and a back-trajectory methodology to provide time- and altitude-dependent volcanic SO2 emission rates for their transport simulations. The authors make good use of AIRS and MIPAS observations to judge the fidelity of their trajectory based transport simulations, which demonstrate the role of the Asian monsoon anticyclone in steering a small but significant contribution of the Sarachev plume to stratospheric aerosol in the tropics.

The paper is generally well written, and will be a useful addition to the literature on this

subject. I have some minor points of clarification, below, which should be addressed.

p.3, line 23: ref "NASA operational data products" Please clarify what products are referred to (OMI?), and why the SO2 index used here is considered "better".

p.4, line 16: what about Dx in the stratosphere and Dz in the troposphere? I note that Hoffman et al. (2016) uses the same values in the stratosphere and troposphere.

p.4, line 19: do you mean constituent (i.e. SO2) mass here?

p.5, line 16: Units of emission rate are given in kg m-1 s-1. Why m-1, not m-3?

p.5, line 36: split (sp)

p.7, line 15: the ASM anticyclone

p.7, lines 19-20: sentence is unclear. Many more aerosol detections are found N of the subtropical jet between 360-400K.

Figures 8, 9, 10, and 11: the PV (black) and geopotential height (red) contours shown appear to be identical for July 10th, July 20th and July 31st. This would appear to be a mistake. The flow fields should change.

p.8, lines 24-25: I see very few MIPAS aerosol detections south of 30N in Fig. 11 (number of detections above 400K), or for the simulated air parcels (Fig.10) so where is "the increased aerosol in the tropical stratosphere above 400K" the authors referring to here?

p.8, lines 28-37, ref - Figure 13: as above, none of the earlier figures demonstrate Sarachev aerosol in the tropics above 400K. Is this then, the first evidence shown of Sarachev aerosol ascending in the tropics? Maybe Fig.13 should be introduced earlier.

p.9, line 18, ASM anticyclone

---

## Referee Comment (RC2) · Anonymous Referee #2 · 16 Jun 2017

Wu et al. studied the dispersion of volcanic aerosols after Sarychev eruption in 2009. The study mainly uses trajectory model, and observations from AIRS and MIPAS. it shows a fairly good model-observation comparison on the SO2 plumes. Wu et al. suggests a 4% of Sarycheve aerosol goes to the Tropical Stratosphere with the help of ASM circulations. In general the topic is interesting, however many issued need to be resolved.

Main Concern:

1. Is your "4%" significant statistics? What is the confidence interval? How is the 4% of volcanic aerosols from Sarychev compared with local background aerosol? 2. I believe

one of the evidence for your equator-ward is MIPAS aerosol detection number data (Figure 5C). The color scale make it quite difficult to tell the numbers, look like there is a factor of 5+ more detections between May.2009 and May.2010. Is it consistent with your estimate? 4% of volcanic aerosol vs. background? 3. Please also comment on uncertainties/noises from observations (MIPAS). Are the signals are robust? 4. Figure 13, you argue the gap between 2009.10 and 2010.2 is due to temperature perturbation. Are you suggesting $H_2SO_4$-$H_2O$ aerosol gets evaporated? I don't think so, the reason is at such low temp, the vapor pressure is super low. Graves may leads to some evaporation, but I assume a few months gap is not expected. Any other reasons? What is the green in early 2009 and late 2010? Any more analysis/evidence suggest they are actually from Sarychev volcanic aerosols for the period of 2009.7-2010.5? 5. Any chance to expand your study to other high-latitude volcanoes? What is the equator-ward transport sensitive to? E.g. injection latitude, altitude, season, location (in/out of ASM), etc? If you move the location north/south, do you expect to get different higher/lower fraction than 4%? I assume if injection is too close to ASM, then you parcels will be trapped in ASM anticyclone instead of going further south? 6. I assume your trajectory model doesn't have aerosol microphysics. Will coagulation and some loss terms that happened in real atmosphere affect your results (i.e. 4%)?

Minor 1. Define latitude range for tropics in the abstract/beginning of the paper

---

## Referee Comment (RC3) · Anonymous Referee #3 · 19 Jun 2017

Fundamentally, this paper is about transport of a small amount of Sarychev aerosol into the tropics in the summer of 2009. This is not a new result, and that there is an anticyclone over the Asian summer monsoon that could have provided the circulation that did this, but would not do so in winter, is also quite obvious and well-known. I cannot understand the units the authors use. 4% of what? Of what parcels? And is this small amount significant? What is the error associated with the calculation of this number? If more than 95% of the aerosols did NOT go to the tropics, isn't that the conventional standard to prove that they did not go to the tropics?

In other words, what is the error bar on this 4%? What is the mass of aerosols that

went into the tropics? What difference did they make for the radiative forcing and for the climate response? How important was it? The discussion on p. 9 is confusing. The authors claim that there was 0.07 Tg S added, but to what? And they claim that there is a 4-7% annual increase, but of what and where? They spelled Hofmann wrong. And clearly this value is not a long-term amount, and Hofmann et al. measured it for a short period of time and a different period than Sarychev.

The paper has a large number of errors and confusing statements. And I am not sure the results are new or interesting. I recommend that the paper either be rejected or sent back to the authors for major revision, clearing up all these issues.

In section 4.2 there are many results about the circulation presented, but the authors never say where they got the data and how they did the calculations. What models or reanalyses were used. The technique of using parcels and trajectories is also quite confusing and not clearly explained.

The authors should use continuous numbering of lines for the entire, and number every one. It is much harder for the reviewer to use page number and line number, and count the numbers and scroll to the top or bottom of the page to find out which line. This annoys reviewers. Make it easier for them!

p. 3, lines 27-28: Emission rates of what? I don't understand how you can use measurements of upper tropospheric SO2 to measure emission rates of SO2 from a volcanic eruption. You have to know what is happening at the volcanic vent, and sample it more often than once or twice a day. And the units in the figure do not make sense to me. You have to explain this.

Even though the emission calculation is explained at the top of p. 5, I do not understand. You have fixed the number of parcels at 100,000 in each column. So how can the number be proportional to the SO2 amount?

The authors seem to confuse emission and injection. Emission is from the volcanic

vent. Injection is where the emissions end up in the atmosphere. They way the paper is written, discussion of emissions and emission rates is very confusing.

There are a number of acronyms that are never defined, like UTLS, TTL, and AIRS.

The paper mixes UK and US spelling. Choose one and use it consistently.

The references are not in alphabetical order.

I do not understand Fig. 2. How can there be emissions of $SO_2$ in the stratosphere? And I also do not understand the units used for emission.

The figures have many errors, with missing sources of the data, missing units, and confusing labeling. See the comments on the attached annotated manuscript.

For example, for Fig. 10 since the unit is % for many figures, yet the shading goes from 0 to 0.007, does this mean all the numbers are « 0.01 %. This is such a small number, why is it even given?

For Fig. 10 and in section 4.2, the data are "above 400 K." What does this mean? There is no temperature above 400 K in the atmosphere. Is this potential temperature? Is it values of potential temperature above 400 K? Is it altitude above the 400 K potential temperature level? The authors have to be very clear with what they mean.

There are multiple English errors.

Please address all of the 101 comments in the attached annotated manuscript.

Please also note the supplement to this comment:
http://www.atmos-chem-phys-discuss.net/acp-2017-425/acp-2017-425-RC3-supplement.pdf

[Figure]

**Supplement:**

[revised manuscript text omitted]

---

## Author Response (AR1)

**Replies to review comments**

We thank the reviewers and the co-editor for the time and effort spent on the manuscript. Please find our point-by-point replies below (in blue color and italics). A revised manuscript with tracked changes will be uploaded.

In the revision of the manuscript, we made the following major changes:

1. The term "$SO_2$ emission rate" in the manuscript was changed into "SO2 emission time series".

2. Equation (1) was added to Section 2.1 to define the AIRS SO2 index used in this paper. Equations (2)-(4) are added to Section 2.3 to explain the MIPAS aerosol detections used in this paper and a brief comment on the MIPAS aerosol data is added too.

3. A brief explanation on the "1.5-day forward and backward trajectories" that we calculated to generate more aerosol data points based on the MIPAS aerosol detections is added in Section 4.2.

4. A brief explanation on the water vapor recorder is added in Section 4.3.

5. A further discussion on the robustness of the percentage of $SO_2$ that entered the tropical stratosphere is added in Section 5.

6. In the discussion and summary, we changed the estimated amount of sulfate aerosol added to the tropical stratosphere to "0.06±0.01 Tg", instead of 0.07 Tg in the original version. This estimate was made based on the "1.2±0.2 Tg" of $SO_2$ injected into the upper troposphere and lower stratosphere (Haywood et al., 2010).

6. The wind vector scale is added in Fig. 1.

7. A dashed line is added in the middle and bottom panels of Fig.5 to show the time of the end of the model simulation. In the bottom panel of Fig. 5, time series of the number of MIPAS aerosol detections above the 380 K and the 400 K isentropic surfaces between 30 °N and 30 °S are added. The explanation of Fig.5 is changed accordingly in Section 4.1.

8. Languages errors have been fixed.

**Anonymous Referee #1**

Wu et al. present a case study of the dispersion of the volcanic plume from the Mt. Sarachev eruption in 2009. They effectively demonstrate the use of AIRS SO2 observations and a back-trajectory methodology to provide time- and altitude-dependent volcanic SO2 emission rates for their transport simulations. The authors make good use of AIRS and MIPAS observations to judge the fidelity of their trajectory based transport simulations, which demonstrate the role of the Asian monsoon anticyclone in steering a small but significant contribution of the Sarachev plume to stratospheric aerosol in the tropics.

The paper is generally well written, and will be a useful addition to the literature on this subject. I have some minor points of clarification, below, which should be addressed.

p.3, line 23: ref "NASA operational data products" Please clarify what products are referred to (OMI?), and why the SO2 index used here is considered "better".

*According to the AIRS Version 6 Release Level 2 Product User Guide, the operational SO2 index in the operational AIRS Level-1B and Level-2 data products uses the channels of 1361.44 cm$^{-1}$ and 1433.06 cm$^{-1}$. The SO$_2$ index used in this study is calculated based on the brightness temperature differences of 1407.21 cm$^{-1}$ and 1371.52 cm$^{-1}$. After comparing with the operational AIRS SO$_2$ index, the SO$_2$ index used here was found to be more sensitive and better at suppressing emissions of interfering species (Hoffmann et al., 2014).*

*We replaced "NASA operational data products" by "AIRS operational SO$_2$ index products" to clarify.*

p.4, line 16: what about Dx in the stratosphere and Dz in the troposphere? I note that Hoffman et al. (2016) uses the same values in the stratosphere and troposphere.

*Dx and Dz is assigned to be 50 m$^2$ s$^{-1}$ and 0 m$^2$ s$^{-1}$ in the troposphere and 0 m$^2$ s$^{-1}$ and 0.1 m$^2$ s$^{-1}$ in the stratosphere. We have made a change in this sentence to clarify.*

p.4, line 19: do you mean constituent (i.e. SO2) mass here?

*Yes, it has been clarified as "the mass of SO$_2$" in this sentence.*

p.5, line 16: Units of emission rate are given in kg m-1 s-1. Why m-1, not m-3?

*In our study, we considered the SO$_2$ emission within a column with a radius of 75 km over the location of Sarychev. So the unit of the SO$_2$ emission kg m$^{-1}$ s$^{-1}$ can be simply converted to kg m$^{-3}$ s$^{-1}$ by dividing by the area. In Fig. 2, we decide to group the SO$_2$ emission only by time and altitude instead of showing the area-averaged value.*

*"kg m$^{-1}$ s$^{-1}$" may not be a reasonable unit for "rate", so we decide to change the "SO$_2$ emission rate" in this paper to "SO$_2$ emission time series".*

p.5, line 36: split (sp)

*Fixed. Thank you.*

p.7, line 15: the ASM anticyclone

*Fixed. Thank you.*

p.7, lines 19-20: sentence is unclear. Many more aerosol detections are found N of the subtropical jet between 360-400K.

*Fixed. Thank you.*

Figures 8, 9, 10, and 11: the PV (black) and geopotential height (red) contours shown appear to be identical for July 10th, July 20th and July 31st. This would appear to be a mistake. The flow fields should change.

*Based on the explanation and calculation method of the PV-based transport barrier (Ploeger et al., 2015), it is not always possible to define a transport barrier on each day or in any month when the Asian summer monsoon is not strong. So we have to show the monthly mean PV-based transport barrier in our figures. That is why the black contours in our figures, which denote the PV-based transport barrier, are identical. And in June, a PV-based transport barrier is not found, so there is no black contour for the top-left panel in Fig.8–11.*

*To make it consistent, we also used the monthly mean geopotential height. So the red contours in our figures are identical too.*

*Thank you for pointing this out. We have added a sentence in the caption of Fig. 8 to clarify.*

p.8, lines 24-25: I see very few MIPAS aerosol detections south of 30N in Fig. 11 (number of detections above 400K), or for the simulated air parcels (Fig.10) so where is "the increased aerosol in the tropical stratosphere above 400K" the authors referring to here?

*Here we refer to the MIPAS aerosol detections with altitude higher than ~18 km shown in the original Fig.5. A sentence has been added here to clarify. And we have revised the Fig.5 in the revised manuscript, so the aerosol above 380 K and above 400 K is shown clearer.*

p.8, lines 28-37, ref - Figure 13: as above, none of the earlier figures demonstrate Sarachev aerosol in the tropics above 400K. Is this then, the first evidence shown of Sarachev aerosol ascending in the tropics? Maybe Fig.13 should be introduced earlier.

*In the revised Fig.5, the altitudes and aerosol and number of MIPAS aerosol detections in the tropical stratosphere above 400K are added. Actually it is in the Fig. 5 was the evidence of Sarychev aerosol in the tropics first shown.*

p.9, line 18, ASM anticyclone

*Fixed. Thank you.*

**Anonymous Referee #2**

Wu et al. studied the dispersion of volcanic aerosols after Sarychev eruption in 2009. The study mainly uses trajectory model, and observations from AIRS and MIPAS. it shows a fairly good model-observation comparison on the SO2 plumes. Wu et al. suggests a 4% of Sarychev aerosol goes to the Tropical Stratosphere with the help of ASM circulations. In general the topic is interesting, however many issued need to be resolved.

Main Concern:

1. Is your "4%" significant statistics? What is the confidence interval? How is the 4% of volcanic aerosols from Sarychev compared with local background aerosol?

*The "4%" is roughly calculated by dividing the number of SO2 parcels between 30 °N and 30 °S above the 380 K isentropic surface by the total number of SO2 parcels released into the upper troposphere and lower stratosphere. A brief explanation has been added in Section 4.1.*

*In the revised the manuscript, we included the time series of the number of MIPAS aerosol detection in the stratosphere between 30 °N and 30 °S in the bottom panel of Fig. 5.These time series show that the number of MIPAS aerosol detections at the end of July 2009, when this "4%" is calculated, is increased by about six times compared with the number before the Sarychev eruption occurred.*

*In the discussion section, we also estimated the mass of sulfate aerosol entering the tropics after the Sarychev eruption. Although the "4%" seems to be small, but the mass of sulfate aerosol is several times larger than the background aerosol.*

2. I believe one of the evidence for your equator-ward is MIPAS aerosol detection number data (Figure 5C). The color scale make it quite difficult to tell the numbers, look like there is a factor of 5+ more detections between May.2009 and May.2010. Is it consistent with your estimate? 4% of volcanic aerosol vs. background?

*We have revised Fig. 5. In the bottom panel, the number of MIPAS aerosol detections in the tropical stratosphere above 380 K and above 400 K is shown. We hope that this revision makes the numbers clear to readers.*

*The "4%" is a proportion to the total number of air parcels we have released in the model simulation, but not to the background aerosol layer. Based on the "4%", we have estimated the mass of sulfate aerosol and the comparison to the background is discussed in the discussion section.*

3. Please also comment on uncertainties/noises from observations (MIPAS). Are the signals are robust?

*Several sentences have been added to Section 2.2 to address the quality of the MIPAS aerosol observations. And the retrieval method is briefly explained in Section 2.2 too.*

*The MIPAS aerosol observations have been used to show the spatial distribution of volcanic aerosols from three strong volcanic eruptions. Two of them are characterized by large SO2 emissions (Grínsvätn, Nabro), and one is characterized with volcanic ash (Puyehue–Cordón Caulle). The MIPAS observations for these three cases were compared with horizontal high-resolution AIRS SO2 and ash index, which verified the capacity of the MIPAS aerosol retrievals in differentiating the aerosol, clear sky, clouds, and ashes (Griessbach et al., 2016).*

4. Figure 13, you argue the gap between 2009.10 and 2010.2 is due to temperature perturbation. Are you suggesting H2SO4-H2O aerosol gets evaporated? I don't think so, the reason is at such low temp, the vapor pressure is super low. Graves may leads to some evaporation, but I assume a few months gap is not expected. Any other reasons? What is the green in early 2009 and late 2010? Any more analysis/evidence suggests they are actually from Sarychev volcanic aerosols for the period of 2009.7-2010.5?

*Actually, the gap is due to an aerosol detection method artefact.*

*The aerosol detection is based on an aerosol-cloud-index (ACI) method described in Griessbach et al. (2016). An analysis of the entire ACI time series shows that there is a very regular gap pattern: every 6 months in about January and July there is a gap. The periodic (semi-annual) changes in the ACI are caused by the radiances in the 960 cm$^{-1}$ window of the AI. At 960 cm$^{-1}$ an impact of CO2 hot bands (at around 50 km) is the most likely explanation for the detection artefact. We added a brief explanation in the revised manuscript.*

5. Any chance to expand your study to other high-latitude volcanoes? What is the equator-ward transport sensitive to? E.g. injection latitude, altitude, season, location (in/out of ASM), etc? If you move the

location north/south, do you expect to get different higher/lower fraction than 4%? I assume if injection is too close to ASM, then you parcels will be trapped in ASM anticyclone instead of going further south?

*The Sarychev case in our study is a reprehensive case to study how a high-latitude eruption can have a notable influence on the tropical and global stratospheric aerosol layer. After this study, it is also a meaningful work to extend our modelling approach and satellite data to other mid- and high-latitude eruptions that have influenced the tropical stratospheric aerosol loading, e.g., Kasatochi (52 °N, August, 2008) and Calbuco (41 °S, April, 2015), to study the efficiency of transport, transport pathway, and further evaluate the role of high-latitude eruption.*

6. I assume your trajectory model doesn't have aerosol microphysics. Will coagulation and some loss terms that happened in real atmosphere affect your results (i.e. 4%)?

*Thank you very much for pointing this out. The MPTRAC model does not resolve aerosol microphysics processes. We admit that meteorological conditions (like wind fields, humidity and temperature) will affect the fate of the SO2 injected into the atmosphere, and of the subsequently formed sulfate aerosols.*

*The "4%" from our model simulation could be affected by many possible mechanisms of the aerosol loss, like the interaction of sulfate aerosol and clouds, and coagulation with other particles. Larger particle size may result in quicker sedimentation rate, especially at higher altitudes where the mean free path between air molecules far exceeds the particle size and particles fall more rapidly than they would otherwise. The scavenging efficiency of SO2 could be increased if it is incorporated into growing ice (Textor et al., 2003). Also, SO2 is slightly soluble in liquid water and it may have a small chance to be washed out during the transport process. But as revealed in our study, the efficient pathway of the transport is approximately between the 360 and 400 K isentropic surfaces, where the atmosphere is relatively dryer, cooler and cleaner than the lower troposphere. So our model results can be considered as an approximate value.*

*We will add this discussion in our manuscript.*

Minor:

1. Define latitude range for tropics in the abstract/beginning of the paper

*Added in the introduction. Thank you.*

**Anonymous Referee #3**

Fundamentally, this paper is about transport of a small amount of Sarychev aerosol into the tropics in the summer of 2009. This is not a new result, and that there is an anticyclone over the Asian summer monsoon that could have provided the circulation that did this, but would not do so in winter, is also quite obvious and well-known.

I cannot understand the units the authors use. 4% of what? Of what parcels? And is this small amount significant? What is the error associated with the calculation of this number? If more than 95% of the aerosols did NOT go to the tropics, isn't that the conventional standard to prove that they did not go to the tropics?

In other words, what is the error bar on this 4%? What is the mass of aerosols that went into the tropics? What difference did they make for the radiative forcing and for the climate response? How important was it? The discussion on p. 9 is confusing. The authors claim that there was 0.07 TgS added, but to what? And they claim that there is a 4-7% annual increase, but of what and where? They spelled Hofmann wrong. And clearly this value is not a long-term amount, and Hofmann et al. measured it for a short period of time and a different period than Sarychev.

*A number of recent studies have shown that the background level of the stratosphere aerosol is increasing after 2000, and the Sarychev eruption is one of the volcanic eruptions that have contributed to this increase (e.g., Solomon et al., 2011). This increase of aerosol in the tropical stratosphere can be found using various satellite data.*

[Figure]

*Figure 1. Time series of monthly mean averaged stratospheric aerosol optical depth (AOD) between 15 and 19 km from OHP LiO3S lidar (top) and time–latitude section of zonal-mean AOD from CALIOP in log-scaled color map with indications of VEI 4 eruptions (bottom). Time periods considered as perturbed by volcanism are shaded light blue in the top panel. White arrows (in 2007–2008) represent the mean meridional component of monthly and zonally averaged horizontal wind at 100 hPa from ERA-Interim reanalysis. Dashed and dotted contours depict the zonal-mean water vapor mixing ratio at 100 hPa from Aura MLS.*

*As shown by the figure from Khaykin et al. (2017), Figure 1, we can see that although the Sarychev peak is located at 48 ˚N, the AOD in the tropical stratosphere was enhanced after its eruption. Just as you mentioned, since the Sarychev eruption happened in summer 2009, one may speculate that the equatorward transport is related to the anticyclonic Rossby wave breaking caused by the Asian summer monsoon (ASM). But one can only assume the ASM helped, but can not answer questions like how did the ASM facilitate the transport? At what vertical range and horizontal location can the ASM help the transport and is there any other factor that can influence the transport? If another high-latitude eruption occurs, can we estimate if the volcanic plume will reach the tropical stratosphere based on information such as the time and location of the eruption and the plume height?*

*In this study, we show that:*

*A high-latitude volcanic eruption can significantly influence the tropical stratosphere under favourable meteorological conditions.*

*The ASM can help to establish transport pathways, but only at specific vertical ranges (360-400 K) and in specific horizontal region (downstream of the ASM anticyclone). The PV-based transport barrier considered in this study marks the boundary of the ASM quite well. It means if the plume is originally outside of the barrier, it may have the chance to be entrained along the circulation and shed to lower latitudes, but if the plume is originally inside of the barrier, it will probably be trapped inside of the ASM.*

*Also, in this paper, we further established a practical method to simulate the time series of the SO2 injected by Sarychev eruption, which is also applicable to other volcanic eruptions. Applying this method to other eruption cases, a time and altitude-resolved volcanic SO2 inventory maybe built up.  This part of the work is relatively new, and may be interesting to scientists who want to enrich the SO2 inventory from volcanisms in their climate model, and scientists who want to compare transport simulations from other Lagrangian trajectory models with ours using the MPTRAC model.*

*As to the specific questions, the "4%" is roughly calculated by dividing the number of SO2 parcels end up in the tropical stratosphere (between 30°N and 30°S above the 380 K isentropic surface ) by the total number of SO2 parcels released into the upper troposphere and lower stratosphere. A brief explanation has been added in Section 4.1.*

*We have stated in our paper that majority of the sulfate aerosol remains at the mid- and high latitudes. However, because the SO2 mass injected by the Sarychev eruption into the atmosphere is large (1.2±0.2Tg), the stratospheric aerosol layer in the tropics is significantly enhanced even though only a small fraction "4%" of the sulfate aerosol entered the tropical stratosphere. In our study, we have shown with MIPAS aerosol detections, that at the end of July 2009 and at the beginning of September 2009, the number of MIPAS aerosol detections in the tropical stratosphere is respectively about seven times and 14 times as large as the number before the Sarychev eruption. We have modified Fig. 5 in our paper to make this enhancement clearer. Figure 1 borrowed from Khaykin et al. (2017) can also demonstrate the enhancement of stratospheric aerosol after the Sarychev eruption.*

*The radiative forcing related to this enhanced aerosol and the climate response are very important to quantify, and they will be part of our future work.*

*If we assume that 4% of 1.4 Tg of SO2 released by Sarychev eruption into the upper troposphere and lower stratosphere is entirely converted into gaseous H2SO4 and entered the tropical stratosphere in a form of 75%-25% H2SO4-H2O solution, this accounts for an additional 0.07 Tg of sulfate aerosol.*

*The "4-7% annual increase" in the background stratospheric aerosol layer is not a result from our study. It is a conclusion in the paper of Hofmann et al. (2009). In Hofmann et al. (2009), the authors used long-term lidars observations at Mauna Loa Observatory in Hawaii (19°N) and Boulder in Colorado (40°N) since 1994 and 2000 respectively. Hofmann et al. (2009) find there is an increasing average trend in aerosol backscatter above 20 km after 2000 of about 4–7% per year, which requires 0.015–0.02 TgS per year to maintain (as in Fig. 2).*

*The Mauna Loa Observatory in Hawaii is located in the tropics, so the stratospheric aerosol data observed there could be a decent long-term record of the tropical stratospheric aerosol. We compare our "0.07 Tg" of sulfate aerosol with this long-term trend to prove the amount of sulfate aerosol added to the tropical stratosphere by the Sarychev eruption is significant comparing with the aerosol background.*

[Figure]

*Figure 2. Integrated backscatter for the 20–25 km altitude range at (a) Mauna Loa Observatory and (b) Boulder, Colorado. (Hofmann et al, 2009)*

The paper has a large number of errors and confusing statements. And I am not sure the results are new or interesting. I recommend that the paper either be rejected or sent back to the authors for major revision, clearing up all these issues.

In section 4.2 there are many results about the circulation presented, but the authors never say where they got the data and how they did the calculations. What models or reanalyses were used. The technique of using parcels and trajectories is also quite confusing and not clearly explained.

*Section 4.2 shows the results we got from forward trajectory simulations carried out with the MPTRAC model. The initialization for each of the parcels is the longitude, latitude, altitude and SO2 mass. The time period covered is from 12 June 2009 to 16 June 2009, when the explosive eruption occurred, as show in Fig. 2. Our initialization is derived by using a backward trajectory method and AIRS SO2 observation, and total SO2 mass estimates from previous studies.*

*Following Hoffmann at al. (2016), 6-hourly ERA-interim wind fields are used to carry out the trajectory simulations. The model outputs are given every six hours, but only results on selected time are shown in Fig. 3 and Fig. 4 for comparison with satellite observations.*

*We believe we have used Section 2.3 to introduce the MPTRAC model and also mentioned the ERA-interim data. Section 3 explains how we carried out the simulation.*

The authors should use continuous numbering of lines for the entire, and number every one. It is much harder for the reviewer to use page number and line number, and count the numbers and scroll to the top or bottom of the page to find out which line. This annoys reviewers. Make it easier for them!

*We changed the line number pattern in our revised version.*

p. 3, lines 27-28: Emission rates of what? I don't understand how you can use measurements of upper

tropospheric SO2 to measure emission rates of SO2 from a volcanic eruption. You have to know what is happening at the volcanic vent, and sample it more often than once or twice a day. And the units in the figure do not make sense to me. You have to explain this.

*Thank you for your suggestion. In fact in this study, the term "SO2 emission" refers to the SO2 emitted or injected into the upper troposphere and lower stratosphere during the explosive volcanic eruption. This clarification has been added to the manuscript where the "SO2 emission" first appears.*

*In our study, we considered the $SO_2$ within a radius of 75 km to the location of Sarychev as the $SO_2$ injected by the volcanic eruption. So the unit of the $SO_2$ emission kg $m^{-1}$ $s^{-1}$ can be simply converted to kg $m^{-3}$ $s^{-1}$ by dividing the area. But the $SO_2$ emission is actually not evenly distributed in this circular area, so we use the exact longitude, latitude, altitude of the $SO_2$ to initialize the trajectories. So in Fig. 2, we group the $SO_2$ emission only by time and altitude instead of showing the area-averaged value.*

Even though the emission calculation is explained at the top of p. 5, I do not understand. You have fixed the number of parcels at 100,000 in each column. So how can the number be proportional to the SO2 amount?

*100,000 air parcels refer to the total number for all AIRS air columns, but the number of air parcels in each of the air columns is different. The air columns that correspond to larger SO2 index (associated with larger SO2 column density) will get more air parcels. For further details, we referred to Hoffmann at al.(2016).*

The authors seem to confuse emission and injection. Emission is from the volcanic vent. Injection is where the emissions end up in the atmosphere. The way the paper is written, discussion of emissions and emission rates is very confusing.

*In the revised manuscript, we changed the "SO2 emission rate" into "SO2 emission time series". The "SO2 emission time series" refers to the mass and altitude of the SO2 emitted or injected into the atmosphere during the volcanic eruption.*

There are a number of acronyms that are never defined, like UTLS, TTL, and AIRS. The paper mixes UK and US spelling. Choose one and use it consistently.

*Fixed. All of the acronyms defined in the abstract are spelled out again the first time they are used in the body of the paper. Thank you.*

The references are not in alphabetical order.

*Fixed. Thank you.*

I do not understand Fig. 2. How can there be emissions of SO2 in the stratosphere? And I also do not understand the units used for emission.

*The study is not concerned with the emission directly at the volcanic vent.*

*As we mentioned above, we will clarify that the "SO2 emission" refers to the SO2 emitted or injected into the atmosphere during the explosive volcanic eruption.*

*The "kg $m^{-1}$ $s^{-1}$" may not be a reasonable unit for "rate", so we change the "$SO_2$ emission rate" in this paper to "$SO_2$ emission time series".*

The figures have many errors, with missing sources of the data, missing units, and confusing labeling. See the comments on the attached annotated manuscript.

*Thank you. Please see the attached manuscript for the replies.*

For example, for Fig. 10 since the unit is % for many figures, yet the shading goes from 0 to 0.007, does this mean all the numbers are «0.01 %. This is such a small number, why is it even given?

*In Fig. 8 and Fig. 10, the shading values are derived by counting the number of air parcels at altitude between the height of the 360 and 400 K isentropic surfaces in each bin and then dividing this number by the total number of air parcels during the simulation time period. The size of the bins is 2 •in longitude ×1 •in latitude. For example, to get the bottom right panel of Fig.8, we first count the total number of the air parcels from 21 to 31 July 2009 (11 days). Since we get the model results four times per day, and the number of air parcels for each model output is 100,000, so the total number X = 100,000×4×11. Then we count the number of air parcels between 360 and 400 K isentropic surfaces in each bin. Assuming this number is Y. The "proportion" equals to Y/X×100.*

*The values are usually very small because the bin size is small (, so the Y is usually not large), but the denominator is very large. However, small values obviously do not equal to meaningless values.*

*We find it is a useful way to show the plume evolution with time. Very similar pictures can be found in other*

*studies, e.g., Fig. 3 in Garny and Randel (2016).*

For Fig. 10 and in section 4.2, the data are "above 400 K." What does this mean? There is no temperature above 400 K in the atmosphere. Is this potential temperature? Is it values of potential temperature above 400 K? Is it altitude above the 400 K potential temperature level? The authors have to be very clear with what they mean.

*Thank you for pointing this out. We mean "altitude above the 400 K potential temperature level"*

There are multiple English errors.

*We have tried to fix the errors in the revision. Once the paper is accepted for final publication, it will undergo language and copy-editing by Copernicus publication.*

Please address all of the 101 comments in the attached annotated manuscript.

*Please see the replies in the attached manuscript.*

Not correct.  VEI is an index of explosivity, not of the impact on climate.  For example, Mount St. Helens was VEI 5, but has no impact on climate.

> Author:     Date: 8/23/2017 10:31:41 PM
> Yes. You are right. But we have to admit that, in the past (even nowadays), when people tried to study the climate impacts of explosive volcanic eruptions, they selected eruption cases by using VEI, instead of erupted SO2 mass. When an increase in the tropical stratospheric aerosol loading since 2000 was found, even without major volcanic eruption, moderate and small volcanic eruption cases defined with smaller VEI got attention.
> Also, in the past, most of the researches on the climate impacts of explosive volcanic eruptions were based on tropical eruptions. But some mid-latitude or high-latitude volcanic eruptions can not only increase the aerosol in the northern hemisphere, but also can enhance the tropical aerosol optical depth and influence the global climate similar to a tropical eruption. However, this influence is conditional. It may require a relatively large amount of SO2 injected and transported to the tropical tropopause layer or higher altitudes and entering the Brewer-Dobson circulation. The time of eruption, the SO2 injection height and the wind fields are all essential elements that determine the transport efficiency. And these are all studied in this paper.
> The St. Hellen eruption in May 1980 you have mentioned could be a very interesting case to compare with the Sarychev case. The St. Hellen eruption did not have impressive global climate impact not only because its relatively small SO2 emission, about 0.775 Tg according to the data provided by the global volcanism program (http://volcano.si.edu/volcano.cfm?vn=321050), but also because of its location and the time of the eruption. St Hellen is a high-latitude volcano and in May 1980, the SO2 was emitted when the meteorological condition was unfavorable of the equatorward transport of the SO2. Quite a lot of volcanic eruptions in the tropics with SO2 injection far less than 0.775 Tg can caused large increase of aerosol optical depth (AOD) in the tropical stratosphere, and subsequently have a great potential to influence the global climate, e.g., Manam (Time: January 2005/Lat: 4.1°S/SO2 mass: 0.14Tg/SO2 injection top height: 24km), Tavurvur (Time: October 2006/Lat: 4.3°S/SO2 mass: 0.3Tg/SO2 injection top height: 18km), and Kelud (Time: February 2014/Lat: 7.9°S/SO2 mass: 0.2Tg/SO2 injection top height: 19km).
> Here in the introduction, we decide to give a very brief description about the related research in the past, and explain why we are interested in the Sarychev case. Although the Sarychev eruption in 2009 was a moderate eruption (in term of VEI), but it is a representative of high-latitude eruptions that contribute to the tropical stratospheric aerosol budget.

**Number: 6**     Author:     Date: 8/23/2017 10:31:41 PM

No.  See previous comment about VEI.  It is the amount of SO2 put into the stratosphere that determines the impact on climate.

> Author:     Date: 8/23/2017 10:31:41 PM
> Please see reply to the previous comment.

**Number: 7**     Author:     Date: 8/23/2017 10:31:41 PM

So why do you even mention VEI above?

> Author:     Date: 8/23/2017 10:31:41 PM
> Please see the reply to the comments above.

**Number: 8**     Author:     Date: 8/23/2017 10:31:41 PM

circulation affects

> Author:     Date: 8/23/2017 10:31:41 PM
> Fixed. Thank you.

**Number: 9**     Author:     Date: 8/23/2017 10:31:41 PM

[revised manuscript text omitted]

**Number: 1**     Author:    Date: 8/23/2017 10:31:41 PM

affects

> Author:    Date: 8/23/2017 10:31:41 PM
> Thank you. But the verb is for "eruptions".

**Number: 2**     Author:    Date: 8/23/2017 10:31:41 PM

If there was such an increase, it was very small. Please explain how large it was, and why you think it was significant.

> Author:    Date: 8/23/2017 10:31:41 PM
> We agree that the influence of Sarychev eruption was not as significant as some other large volcanic eruptions like Pinatubo, or but it clearly increased the topical stratospheric aerosol, which is impressive for a high-latitude eruption. Some figures in previous studies have shown this aerosol increase using Scanning Imaging Absorption Spectrometer for Atmospheric CHartographY (SCIAMACHY), the Optical Spectrograph and InfraRed Imaging System (OSIRIS) and the Cloud-Aerosol Lidar with Orthogonal Polarization (CALIOP) data. We have added two more references here.
> We can see that the increase is very notable. There was not a "number" that we can borrow from previous study to show how large the increase was, but later in this paper, we show with the MIPAS data that the increase is significant and lasted for quite a long time.

**Number: 3**     Author:    Date: 8/23/2017 10:31:41 PM

is

> Author:    Date: 8/23/2017 10:31:41 PM
> Thank you. But the verb is for "eruptions".

**Number: 4**     Author:    Date: 8/23/2017 10:31:41 PM

All acronyms have to be defined. What is this?

> Author:    Date: 8/23/2017 10:31:41 PM
> Fixed. Thank you.

**Number: 5**     Author:    Date: 8/23/2017 10:31:41 PM

brightness temperature differences between what and what?

> Author:    Date: 8/23/2017 10:31:41 PM
> We have added an equation in the manuscript to explain.

**Number: 6**     Author:    Date: 8/23/2017 10:31:41 PM

How can you see emission by looking a the upper troposphere?

> Author:    Date: 8/23/2017 10:31:41 PM
> the $SO_2$ emission in this study refers to the $SO_2$ released or injected into the atmosphere during the process of the explosive eruption. We have add a sentence in the introduction, where the term "emission" first appeared to clarify. Thank you very much for pointing this out.

**Number: 7**     Author:    Date: 8/23/2017 10:31:41 PM

What is this? The temperature of what? If you are using a non-standard index, you have to describe it and explain how it is calculated, and why you used this criterion.

> Author:    Date: 8/23/2017 10:31:41 PM
> Thank you for pointing this out. The equation added above may help to understand. This threshold is provided by previous well-established studies and a reference has been added here. For more details see Hoffmann et al. (2014).

**Number: 8**     Author:    Date: 8/23/2017 10:31:41 PM

Emission of what? Rate of what?

> Author:    Date: 8/23/2017 10:31:41 PM
> Fixed. Thank you.

**Number: 9**     Author:    Date: 8/23/2017 10:31:41 PM

emission of SO2 or of longwave radiation?

> Author:    Date: 8/23/2017 10:31:41 PM
> Here, the "emission" refers to the radiation emission.

[Figure]

mode. On each day, about 14 orbits with about 90 profiles per orbit are measured. From January 2005 to April 2012, the vertical sampling grid spacing between the tangent altitudes is 1.5 km in the UTLS and 3 km above. In this study, we use MIPAS altitude-resolved aerosol data (Griessbach et al., 2016). In the first step, we used the aerosol-cloud-index to identify cloud or aerosol contaminated spectra, and in the second step we use a

5  brightness temperature correction method to separate aerosol from ice clouds. The resulting aerosol product may contain any type of aerosol, e.g. volcanic ash, sulfate aerosol, mineral dust as well as non-ice polar stratospheric clouds (PSCs). MIPAS detected Sarychev aerosol starting on 13 June 2009.

**2.3 MPTRAC**

The Massive-Parallel Trajectory Calculations (MPTRAC) model is a Lagrangian particle dispersion model, which

10  is particularly suited to study volcanic eruption events (Heng et al., 2015; Hoffmann et al., 2016). In the MPTRAC model, trajectories for individual air parcels are calculated based on numerical integration of the kinematic equation of motion and simulations are driven by wind fields from global meteorological reanalyses. Turbulent diffusion is modelled by uncorrelated Gaussian random displacements of the air parcels with zero mean and standard deviations $\sigma_x = \sqrt{D_x \Delta t}$ (horizontally) and $\sigma_z = \sqrt{D_z \Delta t}$ (vertically). $D_x$ and $D_z$ are the

15  horizontal and vertical diffusivities respectively, and $\Delta t$ is the time step for the trajectory calculations. For the Sarychev simulation, $D_x$ is assigned to be 50 $m^2\,s^{-1}$ in the troposphere and $D_z$ is assigned to be 0.1 $m^2\,s^{-1}$ in the stratosphere following Stohl et al. (2005). Furthermore, the sub-grid scale wind fluctuations are simulated by a Markov model (Stohl et al., 2005; Hofmann et al., 2016). Loss processes of chemical species, $SO_2$ in our simulations, are simulated based on an exponential decay of the mass assigned to each air parcel, with a constant

20  half lifetime of seven days.

In this study, the MPTRAC model is driven with the ERA–Interim data (Dee et al., 2011) interpolated on a 1°× 1° horizontal grid on 60 model levels with the vertical range extending from the surface to 0.1 hPa. The ERA–Interim data is provided at 0000, 0600, 1200, and 1800 UTC.

**3 Simulations and observations of the Sarychev plume**

25  ### 3.1 Reconstruction of the Sarychev $SO_2$ emission time series

The Sarychev peak with summit at 1496 m, is located at 48.1°N, 153.2°E, and it is one of the most active volcanoes of the Kuril Islands. It erupted most recently in June 2009 (VEI = 4). On 11 June 2009, two weak ash eruptions were first detected (Levin et al., 2010) and during the main explosive phase from 12 to 16 June 2009, ash, water vapour, and an estimated 1.2±0.2 Tg of $SO_2$ were injected into the UTLS, making it one of the 10

30  largest stratospheric injections in the last 50 years (Haywood et al., 2010). Sulfate aerosol was detected several days after the eruption and the enhancement of the optical depth caused by the Sarychev eruption lasted for months, returning to pre-Sarychev eruption values in the beginning of 2010 (Doeringer et al., 2012; Jégou et al., 2013). As shown in Fig. 1 (top), Sarychev is located at the northern edge of the subtropical jet and to the northeast of the ASM (marked by the black rectangle). In the vertical section (Fig. 1, bottom), the dynamical

35  tropopause, defined by a potential vorticity (PV) value of 2 PV units (PVU), is around 11 km at the location of the Sarychev.

**Page: 4**

[Figure]

**Number: 1**      Author:    Date: 8/23/2017 10:31:41 PM
define.  What is this?

> **Author:**    Date: 8/23/2017 10:31:41 PM
> Fixed. Thank you.

**Number: 2**      Author:    Date: 8/23/2017 10:31:41 PM
What is this?

> **Author:**    Date: 8/23/2017 10:31:41 PM
> Explanations add. Thank you.

**Number: 3**      Author:    Date: 8/23/2017 10:31:41 PM
a Lagrangian

> **Author:**    Date: 8/23/2017 10:31:41 PM
> Fixed. Thank you.

**Number: 4**      Author:    Date: 8/23/2017 10:31:41 PM
what about the stratosphere?

> **Author:**    Date: 8/23/2017 10:31:41 PM
> Fixed. Thank you.

**Number: 5**      Author:    Date: 8/23/2017 10:31:41 PM
what about the troposphere?

> **Author:**    Date: 8/23/2017 10:31:41 PM
> Fixed. Thank you.

**Number: 6**      Author:    Date: 8/23/2017 10:31:41 PM
are

> **Author:**    Date: 8/23/2017 10:31:41 PM
> Fixed. Thank you.

**Number: 7**      Author:    Date: 8/23/2017 10:31:41 PM
Use UK or US spelling, but not both.  If you use sulfate, then use vapor.

> **Author:**    Date: 8/23/2017 10:31:41 PM
> Fixed. Thank you.

**Number: 8**      Author:    Date: 8/23/2017 10:31:41 PM
Why, if it went into the UTLS?  Isn't UT upper troposphere?  How much went into the stratosphere?

> **Author:**    Date: 8/23/2017 10:31:41 PM
> Haywood et al., 2010 got the conclusion by comparing the change of stratospheric aerosol optical depth at 550 nm. Please refer to Haywood et al., 2010 (Table 2).  The Sarychev eruption in 2009 ranks No.9. But they did not estimate the proportion of $SO_2$ that went to the stratosphere. Later in this paper the proportion of $SO_2$ injected into the stratosphere is estimated.

[revised manuscript text omitted]

**Number: 1**       Author:    Date: 8/23/2017 10:31:41 PM
What is this?

> Author:    Date: 8/23/2017 10:31:41 PM
> Please see section 2.1

**Number: 2**       Author:    Date: 8/23/2017 10:31:41 PM
How can this be if you fix the number at 100,000?

> Author:    Date: 8/23/2017 10:31:41 PM
> 100,000 air parcels refer to the total number air parcels in all of the air columns, but the number of air parcels in each of the air columns is different. The air columns that correspond to larger SO2 index (showing larger SO2 column density) will get more air parcels.

**Number: 3**       Author:    Date: 8/23/2017 10:31:41 PM
This

> Author:    Date: 8/23/2017 10:31:41 PM
> Fixed. Thank you.

**Number: 4**       Author:    Date: 8/23/2017 10:31:41 PM
was

> Author:    Date: 8/23/2017 10:31:41 PM
> Fixed. Thank you.

**Number: 5**       Author:    Date: 8/23/2017 10:31:41 PM
is

> Author:    Date: 8/23/2017 10:31:41 PM
> Fixed. Thank you.

**Number: 6**       Author:    Date: 8/23/2017 10:31:41 PM
parcels.

> Author:    Date: 8/23/2017 10:31:41 PM
> Fixed. Thank you.

**Number: 7**       Author:    Date: 8/23/2017 10:31:41 PM
[delete]  Every sentence should be noted or it should not be in the paper.

> Author:    Date: 8/23/2017 10:31:41 PM
> Fixed. Thank you.

**Number: 8**       Author:    Date: 8/23/2017 10:31:41 PM
moved

> Author:    Date: 8/23/2017 10:31:41 PM
> Fixed. Thank you.

[Figure]

diluted and partly depleted and converted into aerosol afterwards and the other two filaments moved toward east. The SO$_2$ plume over the Northwest American continent stretched towards east and west, forming a long filament running through Northern Canada. The SO$_2$ concentration declined exponentially and only a fraction of SO$_2$ remained near the Bering Strait and Northern Canada till 28 June.

5 In Fig. 3, in order to validate the simulation of plume dispersion, and also to indirectly validate the reconstructed emission rates, we compare the simulations of plume dispersion with AIRS SO$_2$ measurements. The SO$_2$ index from AIRS detections was converted into SO$_2$ column density using the correlation function described in Hoffmann et al. (2014), which was built using radiative transfer calculations. AIRS was able to detect the SO$_2$ cloud from the beginning of the eruption of Sarychev up to five weeks later (not fully shown in the figure). The

10 SO$_2$ column density derived from AIRS agrees well with the SO$_2$ column density derived from the Infrared Atmospheric Sounding Interferometer (IASI) in magnitude, e.g. see Fig. 3 in the study of Haywood et al. (2010) and Fig. 2 in the study of Jégou et al. (2013). Generally, the simulations agree well with the AIRS measurements in position and diffusion and the simulations can provide more information on the SO$_2$ distribution than the AIRS measurements alone. The differences between the SO$_2$ clouds, e.g. on 24 June (over the western Pacific)

15 are partly attributed to a mismatch in time of the AIRS SO$_2$ measurements and the simulation output. In magnitude, the SO$_2$ column density from AIRS is slightly larger than that from MPTRAC simulations, and the SO$_2$ maxima are found in different location. This is also found by Hoffmann et al. (2016) for other eruption events and this was attributed to the fact that the inverse modelling approach is optimized to reproduce the spatial extent of the plume but not the absolute emission. Except for discrepancy between the times of the

20 compared data, this remaining difference may also be attributed to the initial setting of the total SO$_2$ mass, the SO$_2$ life time and the uncertainties of the ECMWF-interim winds.

In Fig. 4, the altitudes of the simulated SO$_2$ plume are compared with the tangent altitudes of MIPAS aerosol detections to verify the vertical distribution of the SO$_2$ plume. Aerosol produced by the Sarychev eruption was detected by MIPAS within a few days after the initial eruption. In general, the altitudes of the simulated SO$_2$

25 plume are comparable to the MIPAS aerosol altitudes. The majority of the air parcels were between 10 and 20 km and moved eastwards. A thin filament over the north Pacific ascended to up to 20 km and moved westward to East Asia by the end of June 2009, which was well verified by the MIPAS detections. Some apparent inconsistencies, e.g., along the west coast of North American on 23 June and over the northeast Pacific on 24 and 25 June. We attributed to the fact that the SO$_2$ at lower altitudes (below 14 km) had been converted into

30 aerosol more quickly than the SO$_2$ at higher altitudes (above 16 km). The various vertical distributions of the air parcels were also verified by some overlapping MIPAS detections, e.g., over northwest America. Overall, this comparison demonstrates that the MPTRAC simulation provides a quite accurate representation of the observed horizontal and vertical distribution of the Sarychev SO$_2$ plume.

**4 Equatorward dispersion of the Sarychev plume and the role of the ASM**

35 **4.1 Equatorward dispersion of Sarychev plume**

Although the majority of the Sarychev plume was transported towards the north pole, it's found in our simulations that there was clear equatorward dispersion from extratropical stratosphere to the tropical lower stratosphere, as seen in Fig. 5 (top). The plume reached the tropical stratosphere about one week after the

**Number: 1**     Author:     Date: 8/23/2017 10:31:41 PM
locations.

> Author:     Date: 8/23/2017 10:31:41 PM
> Fixed. Thank you.

**Number: 2**     Author:     Date: 8/23/2017 10:31:41 PM
the discrepancy

> Author:     Date: 8/23/2017 10:31:41 PM
> Fixed. Thank you.

**Number: 3**     Author:     Date: 8/23/2017 10:31:41 PM
ERA

> Author:     Date: 8/23/2017 10:31:41 PM
> Fixed. Thank you.

**Number: 4**     Author:     Date: 8/23/2017 10:31:41 PM
Not a sentence.

> Author:     Date: 8/23/2017 10:31:41 PM
> Fixed. Thank you.

**Number: 5**     Author:     Date: 8/23/2017 10:31:41 PM
America

> Author:     Date: 8/23/2017 10:31:41 PM
> Fixed. Thank you.

**Number: 6**     Author:     Date: 8/23/2017 10:31:41 PM
them

> Author:     Date: 8/23/2017 10:31:41 PM
> Fixed. Thank you.

**Number: 7**     Author:     Date: 8/23/2017 10:31:41 PM
But you are comparing aerosols and not gas.

> Author:     Date: 8/23/2017 10:31:41 PM
> Thank you for pointing this out. We agree it is not accurate to say Fig. 4 (left panels) shows the SO2 plume, although in the beginning of the simulation the plume is composed of SO2.
> The model we have used to simulate the evolution of the plume is not able to simulate the oxidation of SO2. In fact the conversion from SO2 to sulfate aerosol is happening during the transport process. We assume in the validation of the transport simulations, that the SO2 is collocated with the sulfate aerosol, assuming that the sedimentation of the small sulfate aerosol particles is negligible for the time scale considered.

[revised manuscript text omitted]

**T** Number: 1          Author:     Date: 8/23/2017 10:31:41 PM

Increasing from what?  Obviously, if you start at 0, the number increases.

> Author:     Date: 8/23/2017 10:31:41 PM
> The sentences in this paragraph are rephrased.

**T** Number: 2          Author:     Date: 8/23/2017 10:31:41 PM

Where did these data come from?  There are no direct observations of PV.

> Author:     Date: 8/23/2017 10:31:41 PM
> Fixed in the figure caption. Thank you.

**T** Number: 3          Author:     Date: 8/23/2017 10:31:41 PM

units?

> Author:     Date: 8/23/2017 10:31:41 PM
> Fixed. Thank you.

**T** Number: 4          Author:     Date: 8/23/2017 10:31:41 PM

What is this?  Define all acronyms.

> Author:     Date: 8/23/2017 10:31:41 PM
> Fixed. Thank you.

**T** Number: 5          Author:     Date: 8/23/2017 10:31:41 PM

its

> Author:     Date: 8/23/2017 10:31:41 PM
> Fixed. Thank you.

**T** Number: 6          Author:     Date: 8/23/2017 10:31:41 PM

delete

> Author:     Date: 8/23/2017 10:31:41 PM
> Fixed. Thank you.

[Figure]

height criterion. The northern barrier of the subtropical jet was strong, while the southern barrier was more permeable for meridional transport.

This meridional transport under the influence of the ASM revealed by the simulations shown in Fig. 8 is verified by the MIPAS aerosol detections shown in Fig. 9. Because of the sparse horizontal coverage of MIPAS detections, 1.5-day forward and 1.5-day backward trajectories initialized by MIPAS aerosol detections were calculated to fill gaps in space and time. Compared with MIPAS detections in Fig. 9, the simulations in Fig. 8 could successfully reproduce the maxima, minima and filaments of the aerosol distributions. It is also verified by MIPAS detections that the transport barrier of ASM is better demarcated by the PV-based barrier.

The transport pathway to low latitudes for aerosols above 400 K is shown by simulations in Fig. 10 and MIPAS aerosol detections in Fig. 11. At altitudes above the ASM, trajectories are driven by easterlies and meridional, isentropic winds. Till end of July 2009, air parcels above 400 K were transported to lower latitudes (as far as about 15°N), but could not reach the Equator. This suggests that ASM play the most significant role between 360 and 400 K, which may vary in spatial extent associated with the strength of the ASM.

However, a sensitivity test simulation of the same eruption in winter (January 2009) shows that, in northern hemisphere winter, meridional transport from extra-tropic to the tropics is typically suppressed by the strong winter subtropical jets. Forward trajectories initialized by the Sarychev $SO_2$ emissions (as shown in Fig. 2), but driven by winter wind fields in January and February 2009 are shown in Fig. 12. Compared with the summer scenario in Fig. 8, wind speeds in winter 2009 between 360 and 400 K were faster, and the trajectories could span the northern hemisphere within 20 days. About 40 days after eruption, air parcels were almost evenly distributed at mid- and high latitudes, but no air parcels did approach the Equator.

**4.3 Upward transport of Sarychev aerosol**

Simulation results from Figs. 8 to 11 have shown that the ASM anticyclone enhanced the equatorward dispersion of the Sarychev aerosol in the vertical range of 360–400 K, but the ASM did not facilitate the equatorward dispersion above 400K. The increased aerosol in the tropical stratosphere above 400 K (~ 18 km) could only be explained by the upward transport above the TTL. Above zero diabatic heating surface (generally around 360 K), air masses that enter the TTL are considered to be lifted effectively by the radiative heating (Gettelman et al., 2004; Fueglistaler et al., 2009).

In Fig. 13, the upward transport of aerosol in the tropics is clearly demonstrated by the MIPAS aerosol detections at high altitudes between 10°N and 10°S. At 15 km, an enhanced aerosol signal was found from July 2009 and lasted for about 10 months. It returned to pre-Sarychev level at the end of May 2010. At 20 km, enhanced aerosol signal was detected in October 2009, and in another four months, the enhanced aerosol signal was found at 25 km. The overlaid ascent speed of water vapour tape recorder is derived with Aura Microwave Limb Sounder (MLS) measurements from Glanville et al. (2017). The ascent speed of the Sarychev aerosol agrees well with the ascent speed of the water vapour tape recorder before and after the data gap at the end of 2009 and beginning of 2010, which can be explained since the Sarychev aerosol is mainly in form of $H_2SO_4$-$H_2O$ solution. This gap in the MIPAS aerosol data is caused by semi-annual temperature variations at higher altitudes (Griessbach et al., 2016).

**T** Number: 1          Author:     Date: 8/23/2017 10:31:41 PM

Explain in detail what this means and how you did the calculations.

Author:     Date: 8/23/2017 10:31:41 PM
The trajectories are initialized with MIPAS aerosol detections on each day from 12 June 2009 to 31 July 2009. These detections are traced both forward and backward for 1.5 days with the MPTRAC model, so the length of each of the trajectories would be 3 days/72 hours. The model output interval are 6 hours, so this calculation produces aerosol data 12 times as many as the original MIPAS detections.
A brief explanation is added in the manuscript.

**T** Number: 2          Author:     Date: 8/23/2017 10:31:41 PM

above in temperature or altitude?

Author:     Date: 8/23/2017 10:31:41 PM
Fixed. Thank you.

**T** Number: 3          Author:     Date: 8/23/2017 10:31:41 PM

Until the end

Author:     Date: 8/23/2017 10:31:41 PM
Fixed. Thank you.

**T** Number: 4          Author:     Date: 8/23/2017 10:31:41 PM

the zero

Author:     Date: 8/23/2017 10:31:41 PM
Fixed. Thank you.

**T** Number: 5          Author:     Date: 8/23/2017 10:31:41 PM

delete

Author:     Date: 8/23/2017 10:31:41 PM
Fixed. Thank you.

[Figure]

**5 Discussion**

[revised manuscript text omitted]

**Number: 1**      Author:    Date: 8/23/2017 10:31:41 PM

the above

> Author:    Date: 8/23/2017 10:31:41 PM
> Fixed. Thank you.

**Number: 2**      Author:    Date: 8/23/2017 10:31:41 PM

from the

> Author:    Date: 8/23/2017 10:31:41 PM
> Fixed. Thank you.

**Number: 3**      Author:    Date: 8/23/2017 10:31:41 PM

Added to what?

> Author:    Date: 8/23/2017 10:31:41 PM
> Fixed. Thank you.

**Number: 4**      Author:    Date: 8/23/2017 10:31:41 PM

Hofmann

> Author:    Date: 8/23/2017 10:31:41 PM
> Fixed. Thank you.

**Number: 5**      Author:    Date: 8/23/2017 10:31:41 PM

more than what?  More in time or space?

> Author:    Date: 8/23/2017 10:31:41 PM
> Fixed. Thank you.

**Number: 6**      Author:    Date: 8/23/2017 10:31:41 PM

Wrong!  CCN are important in the troposphere, but not in the stratosphere.

> Author:    Date: 8/23/2017 10:31:41 PM
> Here we have deleted several sentences that are not very closely related to our results.
>
> We believe with no doubt that CCN is very important in the troposphere. Here, we did not argue exclusively about the importance of CCN in the stratosphere. Stratospheric aerosols can also sediment into the troposphere, modify the aerosol composition and thus might for example impact cirrus clouds.
> And in fact, CCN also plays a very important role in the stratosphere. Although the stratosphere is much cleaner and dryer than the troposphere, there is a long history of observing the stratospheric clouds. Needless to say, stratospheric clouds have an unignorable effect on the global climate. A significant amount of particulate matter in the stratosphere can act as CCN, e.g. the debris from meteoritic ablation, aluminium oxide spherules from space shuttle launches, and most importantly, the sufate aerosol. For example, the sulfate aerosols in supercooled liquid state or ice particle state provide an essential condition for the formation of stratospheric clouds. The permanent stratospheric sulfate aerosol layer between about 12 and 30 km will be significantly enhanced after explosive volcanic eruptions.
> Reference:  Hamill, P., and Toon, O. B.: Polar stratospheric clouds and the ozone hole, Phys. Today,  44, 34-42, doi: 10.1063/1.881277, 1991.
> Tabazadeh, A., Turco, R. P., Drdla, K., Jacobson, M. Z., and Toon, O. B.: A study of type I polar stratospheric cloud formation, Geophys. Res. Lett., 21, 1619-1622, doi: 10.1029/94GL01368, 1994.

[Figure]

Since the potential climate impact of high-latitude volcanic emissions largely depends on their plume height and the meteorological background consitions, it is essential to initialize
 the simulations with realistic time- and altitude-resolved $SO_2$ emission rate. The backward trajectory approach used in this study to reconstruct the emission time series proves to be an efficient way to produce realistic $SO_2$ emission rate time series.

5   **6 Summary**

In this study, we analysed the equatorward transport pathway of volcanic aerosol from the high-latitude volcanic eruption of the Sarychev in 2009. The analysis was based on MIPAS aerosol detections, AIRS $SO_2$ measurements and trajectory simulations.

First, the time- and altitude-resolved $SO_2$ emission rate was derived using backward trajectories initialized with

10   AIRS $SO_2$ measurements. Second, the dispersion of Sarychev plume from the beginning of the eruption (12 June 2009) to 31 July 2009 was simulated based on the derived $SO_2$ emissions rate. The horizontal distribution of the plume and its altitudes were validated with AIRS $SO_2$ measurements and MIPAS aerosol measurements respectively. The comparisons showed that there was good agreement between the simulations and observations. The results presented in this study suggest that in boreal summer, the transport and dispersion of volcanic

15   emissions were greatly influenced by the dominating ASM circulation, which facilitates the meridional transport of aerosols from the extra-tropical UTLS to the TTL by entraining air along the anticyclonic flows and shedding the air to the deep tropics downstream of the anticyclonic circulation. Meanwhile, the ASM anticyclone effectively isolates the air inside of the ASM from aerosol-rich air outside the anticyclone. The transport barrier at the boundary of the ASM is better denoted by the barrier defined by the PV gradient approach. However, the

20   ASM only influenced the plume dispersion significantly in the vertical range of 360–400 K. The ASM anticyclone did not have notable impact on the equatorward dispersion above 400 K, where the Sarychev aerosol was confined in the northern hemisphere.

The simulations show that until the end of July 2009, about 4% of the emission had entered the tropical stratosphere. Increased number of aerosol was detected by MIPAS in the tropical stratosphere afterwards. After

25   entering the TTL, the aerosol experienced large-scale ascent in the Brewer–Dobson circulation. The ascent speed agrees well with the ascent speed of the water vapour tape recorder. Aerosol signal in the tropics was enhanced within one month of the eruption, and returned to pre-Sarychev level after about 10 months.

The Sarychev eruption had the chance to contribute to the stratospheric aerosol loading in the tropics because it occurred in boreal summer when the equatorward transport was enhanced by the ASM anticyclone. If the

30   eruption happened in winter, the volcanic aerosol would be confined to the latitudes of strong subtropical jets.

**7 Code and data availability**

AIRS data are distributed by the NASA Goddard Earth Sciences Data Information and Services Center (AIRS Science Team and Chahine, 2007). Envisat MIPAS Level-1B data are distributed by the European Space Agency. The ERA–Interim reanalysis data were obtained from the European Centre for Medium-Range

35   Weather Forecasts. The code of the Massive-Parallel Trajectory Calculations (MPTRAC) model is available under the terms and conditions of the GNU General Public License, Version 3 from the repository at https://github.com/slcs-jsc/mptrac (last access: 12 April 2017).

Page: 10

Number: 1    Author:    Date: 8/23/2017 10:31:41 PM
What simulations?

Author:    Date: 8/23/2017 10:31:41 PM
"to initialize the simulations" is changed into "to conduct Lagrangian transport simulations".

Number: 2    Author:    Date: 8/23/2017 10:31:41 PM
wrong English

Author:    Date: 8/23/2017 10:31:41 PM
The sentences here are rephrased.

[revised manuscript text omitted]

**Number: 1**  Author:  Date: 8/23/2017 10:31:41 PM

What is the source for these data?

> Author:  Date: 8/23/2017 10:31:41 PM
> Data source is added in the figure caption. Thank you.

**Number: 2**  Author:  Date: 8/23/2017 10:31:41 PM

what is the scale for the length of the vectors?

> Author:  Date: 8/23/2017 10:31:41 PM
> The scale is added in the top panel. Thank you.

**Number: 3**  Author:  Date: 8/23/2017 10:31:41 PM

indicates

> Author:  Date: 8/23/2017 10:31:41 PM
> Fixed. Thank you.

**Number: 4**  Author:  Date: 8/23/2017 10:31:41 PM

the developing

> Author:  Date: 8/23/2017 10:31:41 PM
> Fixed. Thank you.

**Number: 5**  Author:  Date: 8/23/2017 10:31:41 PM

What is the source of the data?  Why that latitude?  Is this the same time as the top panel?

> Author:  Date: 8/23/2017 10:31:41 PM
> Data source is added in the figure caption.
> The location of the volcano Sarychev peak is 48°N, 153°E, so a vertical line may help readers to know where the volcano is.
> The top panel shows the wind fields from a horizontal view, and the location of the volcano is marked with a red triangle.

[Figure]

[Figure]

[Figure]

**Figure 2: Sarychev SO$_2$ emission time series derived with AIRS measurements using a backward trajectory approach. The emission data is binned every 1 hour and 0.2 km. Black dots denote the height of the thermal tropopause.**

[Figure]
 Number: 1          Author:     Date: 8/23/2017 10:31:41 PM
The units do not seem right.

And how can there be emissions in the stratosphere?  Emissions come from the volcano.

Author:     Date: 8/23/2017 10:31:41 PM
We consider emissions into a vertical column over the volcano, so the unit "kg m-1 s-1" is a reasonable unit. For details please see the point-to-point reply.
The injection height of SO2 by explosive volcanic eruptions can up to more than 10 km, and it is not rare to find SO2 in the stratosphere directly injected by vocalism.

[Figure]

[Figure]

Figure 3: The evolution of $SO_2$ column density from MPTRAC (left, in Dobson units) and $SO_2$ column density from AIRS (right) for the period 16–28 June 2009. The MPTRAC $SO_2$ column density are shown for 0 UTC on selected days and AIRS data are collected within ± 6 hours. Only values larger than 2 DU are shown. The red triangle denotes the location of the Sarychev peak.

Number: 1     Author:     Date: 8/23/2017 10:31:41 PM
DU are not units of density.  This it the column burden.

Author:     Date: 8/23/2017 10:31:41 PM
We believe Dobson units are often used to describe column densities of SO2. We did not mean the "density of SO2 gas" in the caption of Fig. 3, or anywhere else in this paper.

Number: 2     Author:     Date: 8/23/2017 10:31:41 PM
Explain what MPTRAC is and how the data were generated.  Is this a model?  Was it initialized only once at the time of the eruption, or again each day?  From what observations?

Author:     Date: 8/23/2017 10:31:41 PM
Yes, MPTRAC is a model. We have introduced the model in Section 2.3.
The forward trajectories are initialized with air parcels. The essential information for each of the parcels is the longitude, latitude, altitude and SO2 mass. The time period when we have this necessary information is from 12 June 2009 to 16 June 2009, when the explosive eruption occurred, as show in Fig. 2.  This initial information is derived by using a backward trajectory method and the AIRS SO2 observation, and SO2 mass from previous studies.
The forward trajectories are initialized once, and 6-hourly ERA-interim wind fields are used to carry on the trajectories simulations. The model outputs are given every six hours, but only results on selected time are shown in Fig. 3 and Fig. 4 for comparison with satellite observations. Although the simulation was only initialized once, but the simulation results agree well with observations.
Please see section 2.3 and section 3 for introduction of the MPTRAC model and how we carry out the simulation.

[Figure]

[Figure]

[Figure]

**Figure 4: The evolution of SO$_2$ air parcel altitudes from MPTRAC (shown for 0 UTC on selected days) and
[Figure]
 IPAS aerosol detections within ± 6 hours, denoted with color-filled circle markers. The altitudes of all air parcels, regardless of their SO$_2$ values, are shown. The red triangle denotes the location of the Sarychev peak.**

Number: 1    Author:    Date: 8/23/2017 10:31:41 PM
So what is the shading in the MIPAS panels, if the data are only shown in the circles?

Author:    Date: 8/23/2017 10:31:41 PM
MIPAS data is shown with circles and filled with color.

[Figure]

[Figure]

**Figure 5: Top:** percentage (%) of air parcels above 380 K **from MPTRAC simulation binned every 12 hours and 2°
in latitude. Red dots denote the percentage of air parcels between 30°N and 30°S. Middle: MIPAS aerosol altitudes
(km) above 380K from May 2009 to April 2010. Bottom: number of MIPAS aerosol detections. Detections are
binned every 5 days and 5° in latitude. The red triangle denotes the time and latitude of the first Sarychev eruption.**

**T** Number: 1        Author:     Date: 8/23/2017 10:31:41 PM

Is the shading really %?  Does this mean that at every latitude, the value is << 1%?

[Figure]
Author:     Date: 8/23/2017 10:31:41 PM

Yes, in the top panel of Fig. 5, the shading values are the time-latitude distribution of air parcels above the height of 380K in percentage of total number of air parcels. The values are derived by counting the number of air parcels at altitude above the height of the 380 K isentropic surface in each bin and then dividing this number by the total number of air parcels during the simulation time period. The size of the bins is 12 hours × 2° in latitude.

The model outputs are given every 6 hours (4 times per day), and the time period of simulation is from 12 June 2009 to 31 July 2009 (50 days). So the total number of the outputs is 4*50=200. The number of air parcels is assigned to 100,000. So the total number of air parcels during the simulation period is 200*100,000.

The shading values are very small because the denominator is large. We apologize for making it so confusing.

In the revised Fig. 5, we decide to change the way of calculation. The "total number of air parcels in each bin" replaces  the originally used "total number of air parcels during the simulation time period" as the denominator. The total number of air parcels in each bin is 2*100,000. In this way, the shading values will not depend on the simulation time period or the bin size. Instead of showing the relative distribution of the air parcels, the revised figure could give the specific percentage at a given latitude and time.

**T** Number: 2        Author:     Date: 8/23/2017 10:31:41 PM

Potential temperature above 380 K or altitude about theta = 380K?

Author:     Date: 8/23/2017 10:31:41 PM

It is the altitude where the potential temperature equals to 380 K. We have change it into "above the 380 K isentropic surface" to avoid the ambiguity.

[Figure]

**Figure 6:** Zonal mean V anomaly (shaded) in the Asian monsoon anticyclone (40–120°E) with respect to the zonal mean, averaged over 2009 summer (June-August). Zonal wind (black, solid/dashed positive/negative) is averaged between 40 and 120°E. The first thermal tropopause zonally averaged over 0 - 360°E is shown as dashed white line, averaged over 40 to 120°E as thick black line.

**T** Number: 1          Author:     Date: 8/23/2017 10:31:41 PM

What is the source of the data?

  Author:     Date: 8/23/2017 10:31:41 PM
  Data source added. Thank you.

**T** Number: 2          Author:     Date: 8/23/2017 10:31:41 PM

units?

  Author:     Date: 8/23/2017 10:31:41 PM
  Fixed. Thank you.

**T** Number: 3          Author:     Date: 8/23/2017 10:31:41 PM

what does this mean?

  Author:     Date: 8/23/2017 10:31:41 PM
  Fixed. Thank you.

[Figure]

[Figure]

[Figure]

Figure 7: Number of MIPAS aerosol detections between 40°E and 120°E during June-August 2009 (binned every 2 km in altitude and 2° in latitude). Sparse detections (number of detections in each bin is smaller than 10) are shown with black dots. The tropopause, potential temperature, and zonal wind are 1 me as shown in Fig. 6.

**Number: 1**     Author:     Date: 8/23/2017 10:31:41 PM

the same

Author:     Date: 8/23/2017 10:31:41 PM

Fixed. Thank you.

[Figure]

[Figure]

Figure 8: Percentage (%) of air parcels by 30 June (top left), 10 July (top right), 20 July (bottom left) and 31 July 2009 (bottom right) between 360 and 400 K from PTRAC simulations. Results are binned every 2° in longitude and 1° in latitude. The 3,320 geopotential height (m) at 150 hPa is marked in red and the PV-based barrier at 370 K is marked in black. The red triangle denotes the location of the Sarychev.

**Number: 1**     Author:     Date: 8/23/2017 10:31:41 PM
So all the values are << 0.01%?

Author:     Date: 8/23/2017 10:31:41 PM
In Fig. 8, the shading values are derived by counting the number of air parcels at altitude between the height of the 360 and 400 K isentropic surfaces in each bin and then dividing this number by the total number of air parcels during the simulation time period. The size of the bins is 2° in longitude × 1° in latitude.
Similar to the top panel of Fig. 5, the values are very small because the bin size is small but the denominator is large. The shading values in Fig. 10 and Fig. 12 are also generated with the same method. We find it is a useful way to show the plume evolution with time. Very similar pictures can be found in other studies, e.g., Fig. 3 in Garny and Randel (2016).
Reference: Garny, H., and Randel, W. J.: Transport pathways from the Asian monsoon anticyclone to the stratosphere, Atmos. Chem. Phys., 16, 2703-2718, 10.5194/acp-16-2703-2016, 2016.

**Number: 2**     Author:     Date: 8/23/2017 10:31:41 PM
How carried out?  What was the initialization procedure?

Author:     Date: 8/23/2017 10:31:41 PM
Please refer to section 2.3 and section 3.

**Number: 3**     Author:     Date: 8/23/2017 10:31:41 PM
14,320 m geopotential height

Author:     Date: 8/23/2017 10:31:41 PM
Fixed. Thank you.

**Number: 4**     Author:     Date: 8/23/2017 10:31:41 PM
of

Author:     Date: 8/23/2017 10:31:41 PM
"on 150 hPa" is changed into "on the 150 hPa pressure surface"

**Number: 5**     Author:     Date: 8/23/2017 10:31:41 PM
at

Author:     Date: 8/23/2017 10:31:41 PM
"on 370 K" is changed into "on the 370 K isentropic surface".

**Number: 6**     Author:     Date: 8/23/2017 10:31:41 PM
Sarychev volcano.

Author:     Date: 8/23/2017 10:31:41 PM
Fixed. Thank you.

[Figure]

[Figure]

Figure 9: Number of MIPAS aerosol detections and aerosol data generated with 1.5-day forward and 1.5-day backward trajectory calculation, for 30 June (top left), 10 July (top right), 20 July (bottom left) and 31 July (bottom right) between 360 and 400 K. Results are binned every 4° in longitude and 2° in latitude. The 14,320 geopotential height (m) on 150 hPa is marked in red and the PV-based barrier on 370 K is marked in black. Red triangle denotes the location of Sarychev.

**T** Number: 1          Author:     Date: 8/23/2017 10:31:41 PM

What does this mean?  What are detections?  What are data?

> Author:     Date: 8/23/2017 10:31:41 PM
> Fixed. Thank you.

**T** Number: 2          Author:     Date: 8/23/2017 10:31:41 PM

on  [what year?]

> Author:     Date: 8/23/2017 10:31:41 PM
> Fixed. Thank you.

**T** Number: 3          Author:     Date: 8/23/2017 10:31:41 PM

left),

> Author:     Date: 8/23/2017 10:31:41 PM
> Fixed. Thank you.

**T** Number: 4          Author:     Date: 8/23/2017 10:31:41 PM

14,320 m

> Author:     Date: 8/23/2017 10:31:41 PM
> Fixed. Thank you.

**T** Number: 5          Author:     Date: 8/23/2017 10:31:41 PM

[delete]

> Author:     Date: 8/23/2017 10:31:41 PM
> Fixed. Thank you.

[Figure]

[Figure]

Figure 10: Same as Fig. 8 but for percentage (%) of total air parcels above 400 K.

[Figure]

[Figure]

[Figure]

Figure 11: Same as Fig. 9, but for number of detections above 400 K.

[Figure]

[Figure]

Figure 12: Percentage (%) of air parcels for the wintertime sensitivity study,
[Figure]
 10 January (top left), 20 January (top right), 31 January (bottom left) and 10 February 2009 (bottom right) from an MPTRAC simulation for a hypothetical eruption of the Sarychev between 360 and 400 K. Results are binned every 2° in longitude and 1° in latitude. Red triangle denotes the location of Sarychev.

Number: 1     Author:     Date: 8/23/2017 10:31:41 PM
???

Author:     Date: 8/23/2017 10:31:41 PM
Fixed. Thank you.

[Figure]

[Figure]

**Figure 13: 9-day running median of aerosol-cloud-index (ACI) between 10°N and 10°S from 2009 to 2010. Ascent speed of the water vapour tape recorder (solid line) is derived from Glanville et al., 2016 using MLS observations. Black dashed lines indicate the eruption date and 20 July when the simulations show first substantial transport to 10°N–10°S.**
* * *
**T** Number: 1        Author:    Date: 8/23/2017 10:31:41 PM

what are the units?

> **Author:    Date: 8/23/2017 10:31:41 PM**
> Explanation added in the manuscript. Please refer to section 2.2
* * *
**T** Number: 2        Author:    Date: 8/23/2017 10:31:41 PM

for all longitudes?

> **Author:    Date: 8/23/2017 10:31:41 PM**
> Fixed. Thank you.
* * *
**T** Number: 3        Author:    Date: 8/23/2017 10:31:41 PM

What is the speed?  What are the units?  Why is the black line where it is?

> **Author:    Date: 8/23/2017 10:31:41 PM**
> Water vapor above the tropical tropopause layer is considered as a passive tracer. Concentrations of water vapor in the tropical tropopause layer or the tropical lower stratosphere are influenced by the annual cycle in tropical tropopause temperature and this signal is moved upward by the upward branch of the Brewer-Dobson circulation (BDC), creating the so-called tape recorder. So the speed of the water vapor tape recorder is used to investigate the speed of BDC upwelling in the tropics. The tape recorder signal emerges from time–height plots of zonally mean water vapor in the tropical lower stratosphere. The unit for the speed of the water vapor tape recorder is (vertical distance)/time. The commonly used vertical coordinates are pressure levels (hPa) or altitude (km).
> In Glanville et al. (2017), the tape recorder is derived first by obtaining correlation coefficients between daily data at consecutive altitude levels. The data at the higher altitude levels are then shifted in 1-day increments up to 14 months to find the largest correlation coefficient. A strong correlation between the data at the lower level and the shifted data at the higher level is assumed to follow the tape recorder. More details in the calculation of the tape recorder are described in e.g., Minschwaner et al. (2016).
> Usually, the base of the BDC upwelling in the tropics, or upward branch of the BDC is located around 70 hPa, so in Fig. 13, we show the slope from the altitude around 70 hPa. And we adjust the horizontal location of the black solid line to the time when our simulations show first substantial aerosol transport to 10°N–10°S.
> A brief explanation has been added in Section 4.3.
> Reference:  Minschwaner, K., Su, H., and Jiang, J. H.: The upward branch of the Brewer-Dobson circulation quantified by tropical stratospheric water vapor and carbon monoxide measurements from the Aura Microwave Limb Sounder, J. Geophys. Res., 121, 2790–2804, doi: 10.1002/2015JD023961, 2016.
* * *
**T** Number: 4        Author:    Date: 8/23/2017 10:31:41 PM

what is it?

> **Author:    Date: 8/23/2017 10:31:41 PM**
> Rephrased. Thank you.

---

## Referee Report (RR1)

Review of revision of Wu et al. 2017 (ACP)

In general the reviewer addressed my concerns in the revised manuscript, expect for one item:

*Past review question:*
*4. Figure 13, you argue the gap between 2009.10 and 2010.2 is due to temperature perturbation. Are you suggesting H2SO4-H2O aerosol gets evaporated? I don't think so, the reason is at such low temp, the vapor pressure is super low. Graves may leads to some evaporation, but I assume a few months gap is not expected. Any other reasons? What is the green in early 2009 and late 2010? Any more analysis/evidence suggests they are actually from Sarychev volcanic aerosols for the period of 2009.7-2010.5?*

*Actually, the gap is due to an aerosol detection method artefact.*

*The aerosol detection is based on an aerosol-cloud-index (ACI) method described in Griessbach et al. (2016). An analysis of the entire ACI time series shows that there is a very regular gap pattern: every 6 months in about January and July there is a gap. The periodic (semi-annual) changes in the ACI are caused by the radiances in the 960 $cm^{-1}$ window of the AI. At 960 $cm^{-1}$ an impact of CO2 hot bands (at around 50 km) is the most likely explanation for the detection artefact. We added a brief explanation in the revised manuscript.*

**Seems you didn't fully answer/discuss my question. What is the green in early 2009 and late 2010 on your plot? Are they real aerosol (volcanic? or others) or the artifacts you mentioned in the reply?**
**I am a little surprised that MIPAS can tell aerosol/cloud effectively in TTL (down to 15 km), Are the data really meaningful?**
**For now, I am still not convinced on the tap-recorder behavior. Please provide more analysis or you can drop this section.**

---

## Author Response (AR2)

**Replies to reports:**

We thank all the referees and the co-editor for the second review of the manuscript. Please find our point-by-point replies below (in blue color and italics). A revised manuscript with tracked changes will be uploaded.

In the second revision of the manuscript, we made the following changes:

1. Paragraphs are added in Section 2.2 to further explain the MIPAS aerosol retrievals used in this study.

2. Paragraphs are added in Section 4.3 to further explain the MIPAS aerosol retrievals data gap.

3. Two more references are added.

3. Sentence structure and spelling error corrections.

**Reply to referee report #2**

In general the reviewer addressed my concerns in the revised manuscript, expect for one item:
Past review question:
4. Figure 13, you argue the gap between 2009.10 and 2010.2 is due to temperature perturbation. Are you suggesting H2SO4-H2O aerosol gets evaporated? I don't think so, the reason is at such low temp, the vapor pressure is super low. Graves may leads to some evaporation, but I assume a few months gap is not expected. Any other reasons? What is the green in early 2009 and late 2010? Any more analysis/evidence suggests they are actually from Sarychev volcanic aerosols for the period of 2009.7-2010.5?

*Reply: Actually, the gap is due to an aerosol detection method artefact. The aerosol detection is based on an aerosol-cloud-index (ACI) method described in Griessbach et al. (2016). An analysis of the entire ACI time series shows that there is a very regular gap pattern: every 6 months in about January and July there is a gap. The periodic (semi-annual) changes in the ACI are caused by the radiances in the 960 cm-1 window of the AI. At 960 cm−1 an impact of CO2 hot bands (at around 50 km) is the most likely explanation for the detection artefact. We added a brief explanation in the revised manuscript.*

**Seems you didn't fully answer/discuss my question. What is the green in early 2009 and late 2010 on your plot? Are they real aerosol (volcanic? or others) or the artifacts you mentioned in the reply? I am a little surprised that MIPAS can tell aerosol/cloud effectively in TTL (down to 15 km), Are the data really meaningful? For now, I am still not convinced on the tap-recorder behavior. Please provide more analysis or you can drop this section.**

*Sorry that the reply does not fully answer your concern.*

*Based on the retrieval method by Griessbach et al. (2016), smaller ACI values indicate higher aerosol loadings and larger ACI values indicate lower loadings. When studying Section 4.1 and 4.2, aerosol retrievals with ACI < 7 are used. ACI < 7 can cover infrared extinction coefficients as small as $1\times10^{-4} km^{-1}$, which corresponds to $3\times10^{-4} km^{-1}$ in the visible wavelength range (Griessbach et al. 2016). For studying the upward transport in the tropics (Section 4.3), we consider higher ACI values to allow for smaller extinction coefficients.*

*So the green shades show aerosols (not gaps) and they represent relatively larger aerosol loading than the reddish shades. The data oscillations are obvious approximately around January and June each year, as marked by the blue rectangles in Fig.1. They are caused by the AI that uses the atmospheric window region between 960 and 961 $cm^{-1}$, which is overlapped with the $CO_2$ laser bands. Due to the semiannual temperature changes at about 50 km (semiannual oscillation), the $CO_2$ radiance contribution to this window region also oscillates. This oscillation is not only visible at higher altitudes but also at altitudes below the temperature changes because the satellite line-of-sight looks through the whole layer.*

*In addition to the aerosol from the Sarychev eruption, the ACI time series also show aerosol from the Kasatochi eruption 2008 and Merpi eruption 2010 (see Fig.1).*

*According to Griessbach et al. 2016, MIPAS can reliably detect aerosol and clouds down to about 10 km in the tropics and even lower in the extratropics. The discrimination between ice clouds and aerosol in the troposphere was performed for each profile individually. So we consider the data in the tropical tropopause region are robust and meaningful.*
*Also, we have compared the "aerosol tape recorder" revealed by our MIPAS aerosol retrievals with other satellite data sources (not shown in the manuscript), e.g., the CALIOP data (Vernier et al, 2009) and the Odin-OSIRIS aerosol extinction (Rieger et al. 2015), as in Fig. 2.*

*Although there are data oscillations in the MIPAS aerosol retrieval, the tape recorder can still be observed and shows similar trend with the Odin-OSIRIS aerosol extinction. Moreover, this could be a very good application of the MIPAS aerosol retrievals. So we decide to keep the figure.*

*But to clarify the ambiguity, we add these paragraphs in the manuscript:*
*In Section 2.2 where the MIPAS aerosol retrieval method is introduced:*

"Both the CI and the ACI are continuous values. Low values indicate cloudy air and high values indicate clear air. For the CI, Sembhi et al. (2012) defined a set of variable (latitude, altitude, and season) thresholds to discriminate between clear and cloudy air. For the ACI, a fixed threshold value of 7 provides comparable results to the CI threshold (Griessbach et al. 2016)."

"To study the equatorward transport of the Sarychev aerosol (Sections 4.1 and 4.2), we only use MIPAS data with ACI < 7 that can cover infrared extinction coefficients as small as $1 \times 10^{-4} km^{-1}$, which corresponds to $3 \times 10^{-4} km^{-1}$ in the visible wavelength range (Griessbach et al. 2016). For studying the upward transport in the tropics (Section 4.3), we consider all MIPAS retrievals to allow for smaller extinction coefficients.
There are semi-annual data oscillations in the aerosol retrievals. This periodic pattern is caused by the AI that uses the atmospheric window region between 960 and 961 $cm^{-1}$. Around this window region, there are $CO_2$ laser bands. Due to the semiannual temperature changes at about 50 km (semiannual oscillation), the $CO_2$ radiance contribution to this window region also oscillates. As this window is generally very clear of other trace gases, this oscillation is not only visible at higher altitudes but also at altitudes below the temperature changes, because the satellite line-of-sight looks through the whole layer."

*In Section 4.3 where the upward transport demonstrated:*

"It should be noted that the green shades in early 2009 and the dark red shades above the green shades at about 20-26 km in late 2009 show aerosol from the Kasatochi eruption. The green shades in late 2010 and early 2011 are aerosol from the Merapi eruption in October-November 2010. "

"The "aerosol tape recorder" shown by MIPAS data here has first been reported by Vernier et al (2009) in their Section 2.4 using Cloud-Aerosol Lidar with Orthogonal Polarization (CALIOP) data. Other satellite sources, e.g., Optical Spectrograph and InfraRed Imager System (OSIRIS) time series, also have shown this upward transport (Rieger et al. 2015), but the upward transport were not discussed in these publications. "

[Figure]

Figure 1: Fig.13 in the manuscript. MIPAS nine-day running median of aerosol-cloud-index (ACI) between 10 ˚N and 10 ˚S (averaged along longitudes) from 2009 to 2010, overlaid with ascent speed of the water vapor tape recorder (solid line) derived with Aura Microwave Limb Sounder (MLS) water vapor mixing ratio from Glanville et al. (2017). Black dashed lines indicate the 12 June 2009 when the eruption started and 20 July 2009 when the simulations show first substantial aerosol transport to 10 ˚N–10 ˚S. The data gap and aerosol signals from other volcanic eruptions than the Sarychev eruption are schematically marked.

[Figure]

Figure 2: OSIRIS 725nm aerosol extinction between 10 °N and 10 °S (unit: $10^{-3}$ km$^{-1}$ , 10-day mean) 
[revised manuscript text omitted]

---

## Author Response (AR3)

Replies to co-editor:

Dear Dr. Schmidt,

Thank you very much for pointing out the spelling mistakes. We have checked the manuscript once again to correct some typos and grammars. Please find the changes tracked manuscript in the following pages.

Best regards,
Xue Wu on behalf of all the authors

[revised manuscript text omitted]